# ON EMPIRICAL COMPARISONS OF OPTIMIZERS FOR DEEP LEARNING

## ABSTRACT

Selecting an optimizer is a central step in the contemporary deep learning pipeline. In this paper, we demonstrate the sensitivity of optimizer comparisons to the hyperparameter tuning protocol. Our findings suggest that the hyperparameter search space may be the single most important factor explaining the rankings obtained by recent empirical comparisons in the literature. In fact, we show that these results can be contradicted when hyperparameter search spaces are changed. As tuning effort grows without bound, more general optimizers should never underperform the ones they can approximate (i.e., Adam should never perform worse than momentum), but recent attempts to compare optimizers either assume these inclusion relationships are not practically relevant or restrict the hyperparameters in ways that break the inclusions. In our experiments, we find that inclusion relationships between optimizers matter in practice and always predict optimizer comparisons. In particular, we find that the popular adaptive gradient methods never underperform momentum or gradient descent. We also report practical tips around tuning often ignored hyperparameters of adaptive gradient methods and raise concerns about fairly benchmarking optimizers for neural network training.

## 1 INTRODUCTION

The optimization algorithm chosen by a deep learning practitioner determines the training speed and the final predictive performance of their model. To date, there is no theory that adequately explains how to make this choice. Instead, our community relies on empirical studies (Wilson et al., 2017) and benchmarking (Schneider et al., 2019). Indeed, it is the de facto standard that papers introducing new optimizers report extensive comparisons across a large number of workloads. Therefore, to maximize scientific progress, we must have confidence in our ability to make empirical comparisons between optimization algorithms.

Although there is no theory guiding us when comparing optimizers, the popular first-order optimizers form a natural inclusion hierarchy. For example, ADAM (Kingma and Ba, 2015) and RMSPROP (Tieleman and Hinton, 2012) can approximately simulate MOMENTUM (Polyak, 1964) if the $\epsilon$ term in the denominator of their parameter updates is allowed to grow very large. However, these relationships may not matter in practice. For example, the settings of ADAM's hyperparameters that allow it to match the performance of MOMENTUM may be too difficult to find (for instance, they may be infinite).

In this paper, we demonstrate two important and interrelated points about empirical comparisons of neural network optimizers. First, we show that inclusion relationships between optimizers actually matter in practice; in our experiments, more general optimizers *never* underperform special cases. Despite conventional wisdom (Wilson et al., 2017; Balles and Hennig, 2017), we find that when carefully tuned, ADAM and other adaptive gradient methods never underperform MOMENTUM or SGD. Second, we demonstrate the sensitivity of optimizer comparisons to the hyperparameter tuning protocol. By comparing to previous experimental evaluations, we show how easy it is to change optimizer rankings on a given workload (model and dataset pair) by changing the hyperparameter tuning protocol, with optimizer rankings stabilizing according to inclusion relationships as we spend more and more effort tuning. Our findings raise serious questions about the practical relevance of conclusions drawn from these sorts of empirical comparisons.

The remainder of this paper is structured as follows. In Section 2, we review related work, focusing on papers that make explicit claims about optimizer comparisons in deep learning and application papers that provide evidence about the tuning protocols of practitioners. We develop our definition of first-order optimizers in Section 3 along with a notion of inclusion relationships between optimizers. We present our experimental results in Section 4. Despite thorny methodological issues over how to avoid biases in comparisons due to search spaces that favor one optimizer over another, we believe that our experimental methodology is an acceptable compromise and has substantial practical relevance. Among other results, we show that the inclusion hierarchy of update rules is almost entirely predictive of optimizer comparisons. In particular, NADAM (Dozat, 2016) achieves the best top-1 validation accuracy on ResNet-50 on ImageNet in our experiments. The 77.1% we obtain with NADAM, although not as good as the 77.6% obtained using learned data augmentation by Cubuk et al. (2018), is better than the best existing published results using any of the more standard pre-processing pipelines (76.5%, due to Goyal et al. (2017) using MOMENTUM).

## 2 BACKGROUND AND RELATED WORK

Our work was inspired by the recent studies of neural network optimizers by Wilson et al. (2017) and Schneider et al. (2019). Wilson et al. (2017) constructed a simple classification problem in which adaptive gradient methods (e.g. ADAM) converge to provably worse solutions than standard gradient methods. However, crucially, their analysis ignored the $\epsilon$ parameter in the denominator of some adaptive gradient methods. Wilson et al. (2017) also presented experiments in which ADAM produced worse validation accuracy than SGD across *all* deep learning workloads considered. However, they only tuned over the learning rate and learning rate decay scheme in their experiments, leaving all other parameters of ADAM at fixed default values. Despite these findings, adaptive gradient methods continue to be popular since the work of Wilson et al. (2017). Schneider et al. (2019) presented a benchmark suite (DEEPOBS) for deep learning optimizers and reported that there was no single best optimizer across the workloads they considered. Yet Schneider et al. (2019) only tuned the learning rate of each optimizer and left all other hyperparameters at some fixed default values.

As we discuss in Section 4.3, the choices of hyperparameter tuning protocols in Wilson et al. (2017) and Schneider et al. (2019) may be the most important factor preventing their results from being relevant to practical choices about which optimizer to use. Hyperparameter tuning is a crucial step of the deep learning pipeline (Bergstra and Bengio, 2012; Snoek et al., 2012; Sutskever et al., 2013; Smith, 2018), so it is critical for papers studying optimizers to match as closely as possible the tuning protocols of an ideal practitioner. Yet, tuning protocols often differ between works studying neural network optimizers and works concerned with training neural networks to solve specific problems.

Recent papers that study or introduce optimization algorithms tend to compare to ADAM and RMSPROP without tuning their respective $\epsilon$ hyperparameters, presumably to simplify their experiments. It is standard to leave $\epsilon$ at the common default value of $10^{-8}$ for ADAM and $10^{-10}$ for RMSPROP (Tieleman and Hinton, 2012; Kingma and Ba, 2015; Dozat, 2016; Balles and Hennig, 2017; Loshchilov and Hutter, 2017; Zou and Shen, 2018; Ma and Yarats, 2018; Bernstein et al., 2018; Chen et al., 2019; Zou et al., 2019). Others do not even report the value of $\epsilon$ used (Balles and Hennig, 2017; Zhang and Mitliagkas, 2017; Keskar and Socher, 2017; Chen et al., 2018; Zhou et al., 2018; Aitchison, 2018; Reddi et al., 2019; Luo et al., 2019). There are exceptions. Zaheer et al. (2018) and Liu et al. (2019) considered $\epsilon$ values orders of magnitude larger than the standard default. However, the experiments in both papers gave only a limited consideration to $\epsilon$, testing at most two values while tuning ADAM. De et al. (2018) is the only work we found that considered a broad range of values for $\epsilon$. Both Zaheer et al. (2018) and De et al. (2018) found that non-default values of $\epsilon$ outperformed the default.

While it is also extremely common in applications to use a default value of $\epsilon$, some notable papers tuned $\epsilon$ and selected values up to eight orders of magnitude away from the common defaults. Szegedy et al. (2016) used $\epsilon = 1$ for RMSPROP; Liu et al. (2019) reported that their results were sensitive to $\epsilon$ and set $\epsilon = 10^{-6}$ for ADAM; Tan et al. (2019) and Tan and Le (2019) set $\epsilon = 10^{-3}$ for RMSPROP, the latter achieving state-of-the-art ImageNet top-1 accuracy. In reinforcement learning, Hessel et al. (2017) set $\epsilon = 1.5 \times 10^{-4}$.

Despite being introduced solely to prevent division by zero[1] , ADAM's $\epsilon$ can be interpreted in ways that suggest the optimal choice is problem-dependent. If ADAM is interpreted as an empirical, diagonal approximation to natural gradient descent (Kingma and Ba, 2015), $\epsilon$ can be viewed as a multi-purpose damping term whose role is to improve the conditioning of the Fisher, in analogy to the approximate second-order method considered by Becker and Le Cun (1988). We can also view $\epsilon$ as setting a trust region radius (Martens and Grosse, 2015; Adolphs et al., 2019) and controlling an interpolation between momentum and diagonal natural gradient descent, by either diminishing or increasing the effect of $v_t$ on the update direction. Under either interpretation, the best value for $\epsilon$ will be problem-dependent and likely benefit from tuning.

## 3 WHAT IS AN OPTIMIZER?

Optimization algorithms are typically defined by their update rule, which is controlled by hyperparameters that determine its behavior (e.g. the learning rate). Consider a differentiable loss function $\ell : \mathbb{R}^d \to \mathbb{R}$ whose vector of first partial derivatives is given by $\nabla \ell(\theta)$ (more generally, $\nabla \ell(\theta)$ might be a stochastic estimate of the true gradient). In our context, $\ell$ represents the loss function computed over an entire dataset by a neural network and $\theta \in \mathbb{R}^d$ represents the vector of model parameters. The optimization problem is to find a point that (at least locally) minimizes $\ell$. First-order iterative methods for this problem (Nesterov, 2018) construct a sequence $\theta_t$ of iterates converging to a local minimum $\theta_\star$ using queries to $\ell$ and $\nabla \ell$. The sequence $\theta_t$ is constructed by an update rule $\mathcal{M}$, which determines the next iterate $\theta_{t+1}$ from the history $H_t$ of previous iterates along with their function and gradient values, $H_t = \{\theta_s, \nabla \ell(\theta_s), \ell(\theta_s)\}_{s=0}^t$, and a setting of hyperparameters $\phi : \mathbb{N} \to \mathbb{R}^n$. Given an initial parameter value $\theta_0 \in \mathbb{R}^d$, the sequence of points visited by an optimizer with update rule $\mathcal{M}$ is given by,

$$\theta_{t+1} = \mathcal{M}(H_t, \phi_t). \tag{1}$$

The stochastic gradient descent algorithm (SGD; Robbins and Monro, 1951) is one of the simplest such methods used for training neural networks. SGD is initialized with $\theta_0 \in \mathbb{R}^d$, and its hyperparameter is a learning rate schedule $\eta : \mathbb{N} \to (0, \infty)$. The SGD update rule is given by $\text{SGD}(H_t, \eta_t) = \theta_t - \eta_t \nabla \ell(\theta_t)$. The MOMENTUM method due to Polyak (1964) generalizes the SGD method by linearly combining the gradient direction with a constant multiple of the previous parameter update. Its hyperparameters are a learning rate schedule $\eta : \mathbb{N} \to (0, \infty)$ and a momentum parameter $\gamma \in [0, \infty)$,

$$\text{MOMENTUM}(H_t, \eta_t, \gamma) = \theta_t - \eta_t \nabla \ell(\theta_t) + \gamma(\theta_t - \theta_{t-1}). \tag{2}$$

There has been an explosion of novel first-order methods in deep learning, all of which fall into this standard first-order scheme. In Table 1 we list the first-order update rules considered in this paper.

The difference between optimizers is entirely captured by the choice of update rule $\mathcal{M}$ and hyperparameters $\phi$. Since the roles of optimizer hyperparameters on neural network loss functions are not well-understood, most practitioners tune a subset of the hyperparameters to maximize performance over a validation set, while leaving some hyperparameters at fixed default values. The choice of which hyperparameters to tune determines an *effective* family of update rules, and this family is the critical object from a practitioners perspective. Thus, in analogy to (overloaded) function declarations in C++, we define an *optimizer* by an update rule "signature," the update rule name together with the free hyperparameter arguments. For example, in this definition MOMENTUM$(\cdot, \eta_t, \gamma)$ is not the same optimizer as MOMENTUM$(\cdot, \eta_t, 0.9)$, because the latter has two free hyperparameters while the former only has one. ADAM with the default $\epsilon$ is "different" from ADAM with tuned $\epsilon$.

### 3.1 THE TAXONOMY OF FIRST-ORDER METHODS

The basic observation of this section is that some optimizers can approximately simulate others (i.e., optimizer A might be able to approximately simulate the trajectory of optimizer B for any particular setting of B's hyperparameters). This is important knowledge because, as a hyperparameter tuning protocol approaches optimality, a more expressive optimizer can never underperform any of its specializations. To capture the concept of one optimizer approximating another, we define the following inclusion relationship between optimizers.

---

[1]TensorFlow currently refers to $\epsilon$ as "a small constant for numerical stability"; https://www.tensorflow.org/versions/r1.15/api_docs/python/tf/train/AdamOptimizer.

Table 1: Update rules considered in this work. SGD is due to Robbins and Monro (1951), MOMENTUM to Polyak (1964), NESTEROV to Nesterov (1983), RMSPROP to Tieleman and Hinton (2012), and NADAM to Dozat (2016). All operations are taken component-wise for vectors. In particular, for $x \in \mathbb{R}^d$, $x^2$ is a component-wise power function.

$\underline{\text{SGD}(H_t, \eta_t)}$

$\theta_{t+1} = \theta_t - \eta_t \nabla \ell(\theta_t)$

$\underline{\text{MOMENTUM}(H_t, \eta_t, \gamma)}$

$v_0 = 0$

$v_{t+1} = \gamma v_t + \nabla \ell(\theta_t)$

$\theta_{t+1} = \theta_t - \eta_t v_{t+1}$

$\underline{\text{NESTEROV}(H_t, \eta_t, \gamma)}$

$v_0 = 0$

$v_{t+1} = \gamma v_t + \nabla \ell(\theta_t)$

$\theta_{t+1} = \theta_t - \eta_t (\gamma v_{t+1} + \nabla \ell(\theta_t))$

$\underline{\text{RMSPROP}(H_t, \eta_t, \gamma, \rho, \epsilon)}$

$v_0 = 1, m_0 = 0$

$v_{t+1} = \rho v_t + (1 - \rho) \nabla \ell(\theta_t)^2$

$m_{t+1} = \gamma m_t + \dfrac{\eta_t}{\sqrt{v_{t+1} + \epsilon}} \nabla \ell(\theta_t)$

$\theta_{t+1} = \theta_t - m_{t+1}$

$\underline{\text{ADAM}(H_t, \alpha_t, \beta_1, \beta_2, \epsilon)}$

$m_0 = 0, v_0 = 0$

$m_{t+1} = \beta_1 m_t + (1 - \beta_1) \nabla \ell(\theta_t)$

$v_{t+1} = \beta_2 v_t + (1 - \beta_2) \nabla \ell(\theta_t)^2$

$b_{t+1} = \dfrac{\sqrt{1 - \beta_2^{t+1}}}{1 - \beta_1^{t+1}}$

$\theta_{t+1} = \theta_t - \alpha_t \dfrac{m_{t+1}}{\sqrt{v_{t+1}} + \epsilon} b_{t+1}$

$\underline{\text{NADAM}(H_t, \alpha_t, \beta_1, \beta_2, \epsilon)}$

$m_0 = 0, v_0 = 0$

$m_{t+1} = \beta_1 m_t + (1 - \beta_1) \nabla \ell(\theta_t)$

$v_{t+1} = \beta_2 v_t + (1 - \beta_2) \nabla \ell(\theta_t)^2$

$b_{t+1} = \dfrac{\sqrt{1 - \beta_2^{t+1}}}{1 - \beta_1^{t+1}}$

$\theta_{t+1} = \theta_t - \alpha_t \dfrac{\beta_1 m_{t+1} + (1 - \beta_1) \nabla \ell(\theta_t)}{\sqrt{v_{t+1}} + \epsilon} b_{t+1}$

**Definition 1** (Inclusion relationship). Let $\mathcal{M}, \mathcal{N}$ be update rules for use in a first-order optimization method. $\mathcal{M}$ is a subset or specialization of $\mathcal{N}$, if for all $\phi : \mathbb{N} \to \mathbb{R}^n$, there exists a sequence $\psi^i : \mathbb{N} \to \mathbb{R}^m$, such that for all $t \in [0, \infty)$ and histories $H_t$,

$$\lim_{i \to \infty} \mathcal{N}(H_t, \psi_t^i) = \mathcal{M}(H_t, \phi_t)$$

This is denoted $\mathcal{M} \subseteq \mathcal{N}$, with equality $\mathcal{M} = \mathcal{N}$ iff $\mathcal{M} \subseteq \mathcal{N}$ and $\mathcal{N} \subseteq \mathcal{M}$.

Evidently SGD $\subseteq$ MOMENTUM, since $\text{SGD}(H_t, \eta_t) = \text{MOMENTUM}(H_t, \eta_t, 0)$. Many well-known optimizers fall naturally into this taxonomy. In particular, we consider RMSPROP with momentum (Tieleman and Hinton, 2012), ADAM (Kingma and Ba, 2015) and NADAM (Dozat, 2016) (see Table 1) and show the following inclusions in the appendix.

$$\text{SGD} \subseteq \text{MOMENTUM} \subseteq \text{RMSPROP}$$
$$\text{SGD} \subseteq \text{MOMENTUM} \subseteq \text{ADAM} \quad\quad (3)$$
$$\text{SGD} \subseteq \text{NESTEROV} \subseteq \text{NADAM}$$

Note, some of these inclusions make use of the flexibility of hyperparameter schedules (dependence of $\psi^i$ on $t$). In particular, to approximate MOMENTUM with ADAM, one needs to choose a learning rate schedule that accounts for ADAM's bias correction.

If two optimizers have an inclusion relationship, the more general optimizer can never be worse with respect to *any* metric of interest, provided the hyperparameters are sufficiently tuned to optimize that metric. Optimally-tuned MOMENTUM cannot underperform optimally-tuned SGD, because setting $\gamma = 0$ in MOMENTUM recovers SGD. However, optimizers with more hyperparameters might be more expensive to tune, so we should have a theoretical or experimental reason for using (or creating) a more general optimizer. For example, MOMENTUM improves local convergence rates over SGD on twice-differentiable functions that are smooth and strongly convex (Polyak, 1964), and

NESTEROV has globally optimal convergence rates within the class of smooth and strongly convex functions (Nesterov, 1983; 2018).

At first glance, the taxonomy of optimizer inclusions appears to resolve many optimizer comparison questions. However, for a deep learning practitioner, there is no guarantee that the inclusion hierarchy is at all meaningful in practice. For example, the hyperparameters that allow ADAM to match or outperform MOMENTUM might not be easily accessible. They might exist only in the limit of very large values, or be so difficult to find that only practitioners with huge computational budgets can hope to discover them. Indeed, empirical studies and conventional wisdom hold that the inclusion hierarchy does not predict optimizer performance for many practical workloads (Wilson et al., 2017; Balles and Hennig, 2017; Schneider et al., 2019). Either these experimental investigations are too limited or the taxonomy of this section is of limited practical interest and provides no guidance about which optimizer to use on a real workload. In the following section we attempt to answer this question experimentally, and show that these inclusion relationships are meaningful in practice.

# 4 EXPERIMENTS

An empirical comparison of optimizers should aim to inform a careful practitioner. Accordingly, we model our protocol on a practitioner that is allowed to vary all optimization hyperparameters for each optimizer (e.g. $\alpha_t$, $\beta_1$, $\beta_2$, $\epsilon$ for ADAM) in addition to a parameterized learning rate decay schedule, in contrast to studies that fix a subset of the optimization hyperparameters to their default values (e.g. Wilson et al., 2017; Schneider et al., 2019). There is no standard method for selecting the values of these hyperparameters, but most practitioners tune at least a subset of the optimization hyperparameters by running a set of trials to maximize performance over the validation set. In our experiments, we run tens to hundreds of individual trials per workload. Given the variety of workloads we consider, this trial budget covers a wide range of computational budgets.

Selecting the hyperparameter search space for each optimizer is a key methodological choice for any empirical comparison of optimizers. Prior studies have attempted to treat each optimizer fairly by using the same search space for all optimizers (e.g. Wilson et al., 2017; Schneider et al., 2019). However, this requires the assumption that similarly-named hyperparameters should take similar values between optimizers, which is not always true. For example, MOMENTUM and NESTEROV both have similar-looking momentum and learning rate hyperparameters, but NESTEROV tolerates larger values of its momentum hyperparameter (Sutskever et al., 2013), so any fixed search space will likely be more favorable for one of the two. The situation worsens with less closely related optimizers, and designing a search space that is equally appropriate for optimizers with incommensurate hyperparameters is almost impossible. Despite coming with its own set of challenges, it is most informative to compare optimizers assuming the practitioner is allowed to tune hyperparameters for different optimizers independently by way of optimizer-specific search spaces.

In our experiments, we chose the search space for each optimizer by running an initial set of experiments over a relatively large search space. In a typical case, we ran a single set of initial trials per optimizer to select the final search space. However, in some cases we chose the initial search space poorly, so we ran another set of experiments to select the final search space. The effort required to choose each search space cannot simply be quantified by the number of initial trials; the provenance of each search space is difficult to trace exactly. In some cases, our search spaces were informed by published results or prior experience with particular models and optimizers. We note that this is true of all search spaces in the literature: they are hard-won treasures that tend to be refined over many experiments and across many workloads, representing the sum total of our community's experience. We validated our search spaces by checking that that the optimal hyperparameter values were away from the search space boundaries for all optimizers in all experiments (see Figure 5 in Appendix E). We provide our final search spaces for all experiments in Appendix D. The fact that our final error rates compare favorably to prior published results – including reaching state-of-the-art for our particular configuration of ResNet-50 on ImageNet (see Section 4.2) – supports our claim that our methodology is highly competitive with expert tuning procedures.

Table 2: Summary of workloads used in experiments.

| Task | Evaluation metric | Model | Dataset | Target error | Batch size | Budget |
|---|---|---|---|---|---|---|
| Image classification | Classification error | Simple CNN | Fashion MNIST | 6.6% | 256 | 10k steps |
| | | ResNet-32 | CIFAR-10 | 7% | 256 | 50k steps |
| | | CNN | CIFAR-100 | – | 256 | 350 epochs |
| | | VGG-16 | CIFAR-10 | – | 128 | 250 epochs |
| | | ResNet-50 | ImageNet | 24% | 1024 | 150k steps |
| Language modeling | Classification error | LSTM | War and Peace | – | 50 | 200 epochs |
| | Cross entropy | Transformer | LM1B | 3.45 | 256 | 750k steps |

## 4.1 OVERVIEW OF WORKLOADS AND EXPERIMENTAL DETAILS

We investigated the relative performance of optimizers across a variety of image classification and language modeling tasks. For image classification, we trained a simple convolutional neural network (Simple CNN) on Fashion MNIST (Xiao et al., 2017); ResNet-32 (He et al., 2016a) on CIFAR-10 (Krizhevsky, 2009); a CNN on CIFAR-100; VGG-16 (Simonyan and Zisserman, 2014) on CIFAR-10; and ResNet-50 on ImageNet (Russakovsky et al., 2015). For language modeling, we trained a 2-layer LSTM model (Hochreiter and Schmidhuber, 1997) on Tolstoy's *War and Peace*; and Transformer (Vaswani et al., 2017) on LM1B (Chelba et al., 2014). We used a linear learning rate decay schedule parameterized the same way as Shallue et al. (2019) for all workloads. We used a fixed batch size and a fixed budget of training steps for each workload independent of the optimizer. Table 2 summarizes these workloads and Appendix B provides the full details.

Given a hypercube-shaped search space, our tuning protocol sought to model a practitioner with a fixed budget of trials trying to achieve the best outcome using tens of feasible trials (10, 50, or 100 depending on the workload).[2] A feasible trial is any trial that achieves finite training loss. We used quasi-random uniform search (Bousquet et al., 2017), and continued the search until we obtained a fixed number of feasible trials. From those trials we considered two statistics. The first, in order to characterize the best outcome, is a metric of interest (e.g. test accuracy) corresponding to the trial achieving the optimum of some other metric (e.g. validation accuracy). The second, in order to characterize the speed of training, is the number of steps required to reach a fixed validation target conditional on at least one trial in the search having reached that target. We chose the target for each workload based on initial experiments and known values from the literature (see Table 2). We estimated means and uncertainties using the bootstrap procedure described in Appendix C.

## 4.2 INCLUSION RELATIONSHIPS MATTER IN PRACTICE

Figure 1 shows the final predictive performance of six optimizers on four different workloads after tuning hyperparameters to minimize validation error. Regardless of whether we compare final validation error or test error, the inclusion relationships hold in all cases – a more general optimizer never underperforms any of its specializations within the error bars. Similar results hold for training error (see Figure 9 in Appendix E). Training speed is also an important consideration, and Figure 2 demonstrates that the inclusion relationships also hold within error bars when we compare the number of steps required to reach a target validation error. Moreover, these results confirming the relevance of optimizer inclusion relationships do not depend on the exact step budgets or error targets we chose (see Figure 10 in Appendix E), although large changes to these values would require new experiments.

---

[2]Although we used a budget of tens of independent tuning trials throughout this section, in retrospect the best validation error across tuning trials converged quite quickly for our final search spaces, producing good results with fewer than 20 trials in many cases. See Figures 6– 8 in Appendix E.

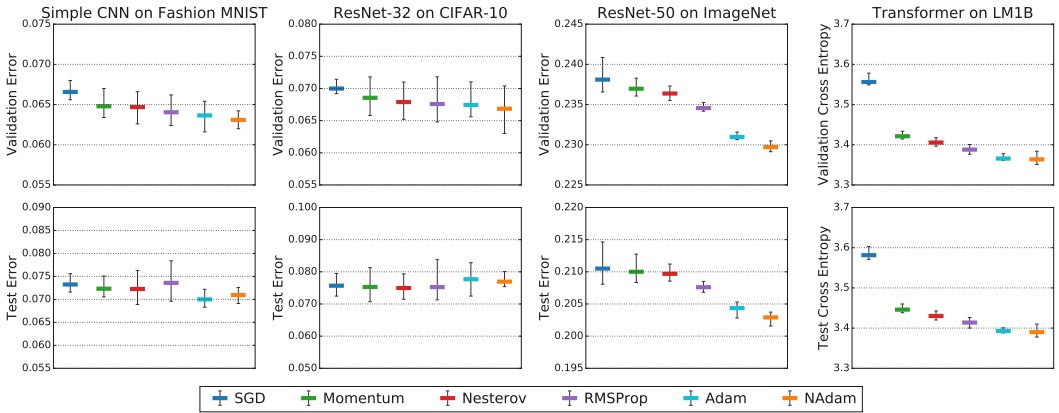

Figure 1: The relative performance of optimizers is consistent with the inclusion relationships, regardless of whether we compare final validation error (top) or test error (bottom). For all workloads, we tuned the hyperparameters of each optimizer separately, and selected the trial that achieved the lowest final validation error.

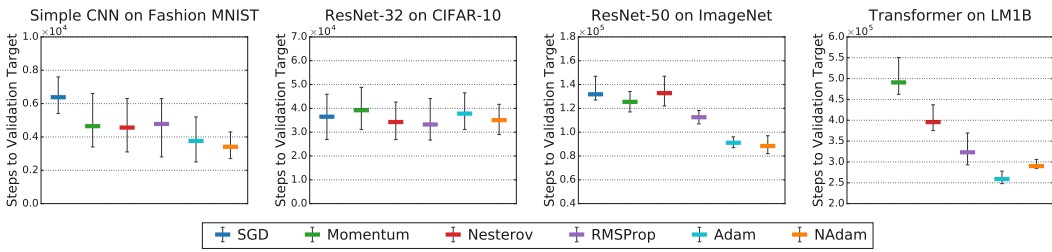

Figure 2: The relative training speed of optimizers is consistent with the inclusion relationships. We measured (idealized) training speed as the number of training steps required to reach a target validation error (see Table 2 for the error targets).

Of course, just because a more general optimizer is no worse than any of its specializations doesn't mean the choice of optimizer makes a large difference on all workloads. For some workloads in Figures 1 and 2, all optimizers perform about the same, while other workloads have a clear ranking or even dramatic differences. For example, the choice of optimizer seems to make little difference for ResNet-32 on CIFAR-10; all optimizers achieve similar predictive performance and training speed. On the other hand, Transformer on LM1B exhibits a clear ranking in terms of predictive performance and training speed. For this workload, ADAM needs roughly half the steps that MOMENTUM requires to reach our target error, and, although not shown in Figure 2, roughly six times fewer steps to get the same result as SGD. These differences are clearly significant enough to matter to a practitioner, and highlight the practical importance of choosing the right optimizer for some workloads.

The most general optimizers we considered were RMSPROP, ADAM, and NADAM, which do not include each other as special cases, and whose relative performance is not predicted by inclusion relationships. Across the workloads we considered, none of these optimizers emerged as the clear winner, although ADAM and NADAM generally seemed to have an edge over RMSPROP. For all of these optimizers, we sometimes had to set the $\epsilon$ parameter orders of magnitude larger than the default value in order to get good results. In particular, we achieved a validation accuracy of 77.1% for ResNet-50 on ImageNet using NADAM with $\epsilon = 9475$, a result that exceeds the 76.5% achieved by Goyal et al. (2017) using MOMENTUM. Across just these 4 workloads, the range of the optimal values of the $\epsilon$ parameter spanned 10 orders of magnitude. Faced with this reality, a practitioner might reasonably doubt their ability to find a value near the optimum. However, we found that we could reasonably expect to find a suitable value with only tens of trials. When tuning $\epsilon$ for ADAM or NADAM over a large range, we found it more efficient to search over $(\epsilon, \alpha_0/\epsilon)$ instead of $(\epsilon, \alpha_0)$; see Appendix D for more details.

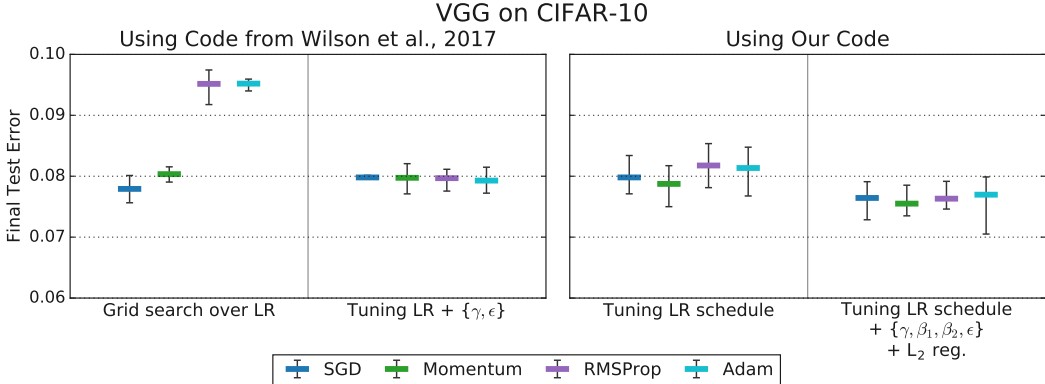

Figure 3: Tuning more hyperparameters removes the differences in test error between optimizers observed by Wilson et al. (2017). Tuning a subset of optimizer hyperparameters and the initial learning rate is sufficient to equalize performance between all optimizers (left). More extensive hyperparameter tuning in our setup, including the learning rate schedule, improves results for all optimizers and still does not produce any differences between optimizer performances (right).

### 4.3 RECONCILING DISAGREEMENTS WITH PREVIOUS WORK

In order to confirm that differences in hyperparameter tuning protocols explain the differences between our conclusions and those of Wilson et al. (2017) and Schneider et al. (2019), we reproduced a representative subset of their results and then inverted, or at least collapsed, the ranking over optimizers just by expanding the hyperparameter search space.

The left pane of Figure 3 shows our experiments on VGG on CIFAR-10 using code released by Wilson et al. (2017). When we match their protocol and perform their grid search over the initial learning rate and no other tuning, we reproduce their original result showing worse test error for RMSPROP and ADAM. However, when we tune the momentum parameter and $\epsilon$ with random search, all four optimizers reach nearly identical test error rates.[3] With our learning rate schedule search space, merely tuning the learning rate schedule was enough to make all optimizers reach the same test error within error bars. When we additionally tuned the optimization hyperparameters and weight decay in our setup we also get similar results for all optimizers, removing any evidence the inclusion relationships might be violated in practice.

Figure 4 shows our results with different tuning protocols for a CNN on CIFAR-100 and an LSTM language model trained on *War and Peace* to match the experiments in Schneider et al. (2019). As reported by Schneider et al. (2019), if we only tune the learning rate without tuning the decay schedule or other optimizer hyperparameters, ADAM does worse than MOMENTUM for the CNN and SGD performs slightly better than ADAM and MOMENTUM on the *War and Peace* dataset, although Schneider et al. (2019) found a larger advantage for SGD. However, once we tune the all the optimizer hyperparameters, ADAM does better than MOMENTUM which does better than SGD, as predicted by the inclusion relationships.

We conclude that the reason both Schneider et al. (2019) and Wilson et al. (2017) observed a ranking that, at first glance, contradicts the inclusion relationships is because they were not tuning enough of the hyperparameters. If we recast their results in our terminology where ADAM with default $\epsilon$ is a different optimizer than ADAM with $\epsilon$ tuned then there is no contradiction with our results and it becomes clear immediately that they do not consider the most interesting form of ADAM for practitioners.

---

[3]Wilson et al. (2017) selected trials to minimize the training loss and then report test set results. As Figure 3 shows, removing this somewhat non-standard choice and tuning on a validation set and reporting test set results does not change anything.

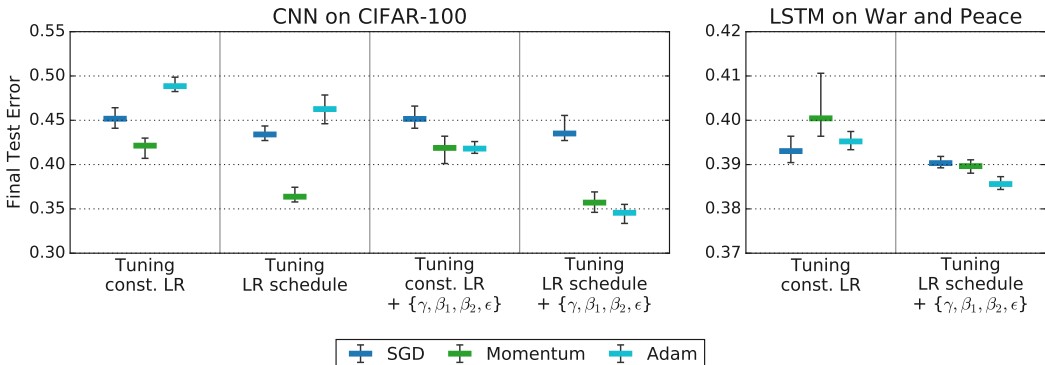

Figure 4: Tuning more hyperparameters changes optimizer rankings from Schneider et al. (2019) to rankings that are consistent with the inclusion relationships. The leftmost columns for each workload reproduce the rankings from Schneider et al. (2019), while the remaining columns tune over increasingly general search spaces. All columns use our random search tuning protocol.

## 5 CONCLUSIONS

Inspired by the recent efforts of Wilson et al. (2017) and Schneider et al. (2019), we set out to provide a detailed empirical characterization of the optimizer selection process in deep learning. Our central finding is that inclusion relationships between optimizers are meaningful in practice. When tuning all available hyperparameters under a realistic protocol at scales common in deep learning, we find that more general optimizers never underperform their special cases. In particular, we found that RMSPROP, ADAM, and NADAM never underperformed SGD, NESTEROV, or MOMENTUM under our most exhaustive tuning protocol. We did not find consistent trends when comparing optimizers that could not approximate each other. We also found workloads for which there was not a statistically significant separation in the optimizer ranking.

Our experiments have some important limitations and we should be careful not to overgeneralize from our results. The first major caveat is that we did not measure the effects of varying the batch size. Recent empirical work (Shallue et al., 2019; Zhang et al., 2019) has shown that increasing the batch size can increase the gaps between training times for different optimizers, with the gap from SGD to MOMENTUM (Shallue et al., 2019) and from MOMENTUM to ADAM (Zhang et al., 2019) increasing with the batch size. Nevertheless, we strongly suspect that the inclusion relations would be predictive at any batch size under a tuning protocol similar to the one we used. The second important caveat of our results is that they inevitably depend on the tuning protocol and workloads that we considered. Although we made every attempt to conduct realistic experiments, we should only expect our detailed findings to hold for similar workloads under similar protocols, namely uniform quasi-random tuning for tens to hundreds of trials, over hypercube search spaces, and with our specific learning rate schedule parameterization. Nevertheless, these caveats reinforce our central point: all empirical comparisons of neural network optimizers depend heavily on the hyperparameter tuning protocol, perhaps far more than we are used to with comparisons between model architectures.

If we were to extract "best practices" from our findings, then we suggest the following. If we can afford tens or more runs of our code, we should tune all of the hyperparameters of the popular adaptive gradient methods. Just because two hyperparameters have a similar role in two different update rules doesn't mean they should take similar values— optimization hyperparameters tend to be coupled and the optimal value for one may depend on how the others are set. Our results also confirm that the optimal value of Adam's $\epsilon$ is problem-dependent, so the onus is on empirical studies that fix $\epsilon = 10^{-8}$ to defend that choice. Finally, we should be skeptical of empirical comparisons of optimizers in papers, especially if an optimizer underperforms any of its specializations. When we do inevitably compare optimizers, we should report search spaces and highlight decisions about what hyperparameters were tuned when interpreting results.

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

## A  OPTIMIZER INCLUSIONS

Table 1 summarizes the update rules for the optimizers we consider in this work. We assume update rules as implemented in TensorFlow r1.15. RMSPROP includes momentum. Here we prove the their inclusion relationships, see Definition 1.

**MOMENTUM can exactly implement SGD**

MOMENTUM$(I_t, \eta_t, 0) = $ SGD$(I_t, \eta_t)$, so SGD $\subseteq$ MOMENTUM.

**NESTEROV can exactly implement SGD**

NESTEROV$(I_t, \eta_t, 0) = $ SGD$(I_t, \eta_t)$, so SGD $\subseteq$ NESTEROV.

**RMSPROP with momentum can exactly implement MOMENTUM**

Consider RMSPROP$(I_t, \eta_t, \gamma, \rho = 1, \epsilon = 0)$, so that

$$m_{t+1} = \gamma m_t + \eta_t \nabla \ell(\theta_t) \,,$$
$$\theta_{t+1} = \theta_t - m_{t+1} \,.$$

This is equivalent to MOMENTUM, since

$$m_{t+1}^{(\text{RMSPROP})} \equiv \eta_t v_{t+1}^{(\text{MOMENTUM})} \,.$$

Thus RMSPROP$(I_t, \eta_t, \gamma, 1, 0) = $ MOMENTUM$(I_t, \eta_t, \gamma)$, so MOMENTUM $\subseteq$ RMSPROP.

**RMSTEROV can exactly implement NESTEROV**

Consider RMSTEROV$(I_t, \eta_t, \gamma, \rho = 1, \epsilon = 0)$, so that

$$m_{t+1} = \gamma m_t + \eta_t \nabla \ell(\theta_t) \,,$$
$$\theta_{t+1} = \theta_t - [\gamma m_{t+1} + \eta_t \nabla \ell(\theta_t)] \,.$$

This is equivalent to MOMENTUM, since

$$m_{t+1}^{(\text{RMSPROP})} \equiv \eta_t v_{t+1}^{(\text{NESTEROV})} \,.$$

Thus RMSTEROV$(I_t, \eta_t, \gamma, 1, 0) = $ MOMENTUM$(I_t, \eta_t, \gamma)$, so MOMENTUM $\subseteq$ RMSTEROV.

**ADAM can approximate MOMENTUM for large $\epsilon$**

Consider ADAM$(I_t, \alpha_t = \epsilon \eta_t (1 - \gamma^t), \beta_1 = \gamma, \beta_2 = 0, \epsilon)$, so that

$$m_{t+1} = \gamma m_t + (1 - \gamma) \nabla \ell(\theta_t) \,,$$
$$\theta_{t+1} = \theta_t - \frac{\eta_t}{(1 - \gamma)} \left[ \frac{m_{t+1}}{|\nabla \ell(\theta_t)|/\epsilon + 1} \right] \,.$$

If $\epsilon$ is large, so that $|\nabla \ell(\theta_t)|/\epsilon \ll 1$, then

$$m_{t+1} = \gamma m_t + (1 - \gamma) \nabla \ell(\theta_t) \,,$$
$$\theta_{t+1} = \theta_t - \eta_t \frac{m_{t+1}}{1 - \gamma} \,.$$

This is equivalent to MOMENTUM, since

$$m_t^{(\text{ADAM})} \equiv (1 - \gamma) \, v_t^{(\text{MOMENTUM})} .$$

Thus $\lim_{\epsilon \to \infty}$ ADAM$(I_t, \epsilon \eta_t (1 - \gamma^t), \gamma, 0, \epsilon) = $ MOMENTUM$(I_t, \eta_t, \gamma)$, so MOMENTUM $\subseteq$ ADAM

**NADAM can approximate NESTEROV for large $\epsilon$**

Consider NADAM$(I_t, \alpha_t = \epsilon \eta_t (1 - \gamma^t), \beta_1 = \gamma, \beta_2 = 0, \epsilon)$, so that

$$m_{t+1} = \gamma m_t + (1 - \gamma) \nabla \ell(\theta_t) \,,$$
$$\theta_{t+1} = \theta_t - \frac{\eta_t}{(1 - \gamma)} \left[ \frac{\gamma m_{t+1} + (1 - \gamma) \nabla \ell(\theta_t)}{|\nabla \ell(\theta_t)|/\epsilon + 1} \right] \,.$$

If $\epsilon$ is large, so that $|\nabla\ell(\theta_t)|/\epsilon \ll 1$, then

$$m_{t+1} = \gamma m_t + (1-\gamma)\nabla\ell(\theta_t),$$
$$\theta_{t+1} = \theta_t - \eta_t \left[ \frac{\gamma m_{t+1}}{1-\gamma} + \nabla\ell(\theta_t) \right].$$

This is equivalent to NESTEROV, since

$$m_t^{(\text{NADAM})} \equiv (1-\gamma)\, v_t^{(\text{NESTEROV})}$$

Thus $\lim_{\epsilon\to\infty} \text{NADAM}(I_t, \epsilon\,\eta_t(1-\gamma^t), \gamma, 0, \epsilon) = \text{NESTEROV}(I_t, \eta_t, \gamma)$, so NESTEROV $\subseteq$ NADAM.

## B  WORKLOAD DETAILS

This section details the datasets and models summarized in Table 2.

### B.1  DATASET DESCRIPTIONS

For Fashion MNIST, CIFAR-10, ImageNet, and LM1B, our setup was identical to Shallue et al. (2019) except for the image pre-processing details described below. For *War and Peace*, our setup was identical to the "Tolstoi" dataset of Schneider et al. (2019).

**CIFAR-10/100:** We pre-processed images by subtracting the average value across all pixels and channels and dividing by the standard deviation.[4] For experiments with the ResNet-32 and CNN models, we followed the standard data augmentation scheme used in He et al. (2016a): 4 pixels padded on each side with single random crop from padded image or its horizontal reflection. We did not use random cropping for experiments with VGG for consistency with Wilson et al. (2017).

**ImageNet:** We augmented images at training time by resizing each image, taking a random crop of $224 \times 224$ pixels, randomly horizontally reflecting the cropped images, and randomly distorting the image colors. At evaluation time, we performed a single central crop of $224 \times 224$ pixels. In both training and evaluation, we then subtracted the global mean RGB value from each pixel using the values computed by Simonyan and Zisserman (2014).[5]

### B.2  MODEL DESCRIPTIONS

**Simple CNN** is identical to the base model described in Shallue et al. (2019). It consists of 2 convolutional layers with max pooling followed by 1 fully connected layer. The convolutional layers use $5 \times 5$ filters with stride 1, "same" padding, and ReLU activation function. Max pooling uses a $2 \times 2$ window with stride 2. Convolutional layers have 32 and 64 filters each and the fully connected layer has 1024 units. It does not use batch normalization.

**CNN** is the "All-CNN-C" model from Springenberg et al. (2014), as used in Schneider et al. (2019). The model consists of 3 convolutional layer blocks with max pooling. The convolutional layers use $5 \times 5$ filters with stride 1, "same" padding, and ReLU activation function. Max pooling uses a $2 \times 2$ window with stride 2. Convolutional layer blocks have 96, 192 and 192 filters each. As in Schneider et al. (2019), we used $L_2$ regularization of $5 \times 10^{-4}$.

**ResNet** is described in He et al. (2016a). We used the improved residual block described in He et al. (2016b). We used batch normalization (Ioffe and Szegedy, 2015) with exponential moving average (EMA) decay of 0.997 for ResNet-32, and ghost batch normalization (Hoffer et al., 2017) with ghost batch size of 32 and EMA decay of 0.9 for ResNet-50.

**VGG** is based on "model C" from Simonyan and Zisserman (2014). It consists of 13 convolutional layers followed by 3 fully connected hidden layers. We followed the modification used by Wilson et al. (2017) with batch normalization layers.

---

[4]We used the TensorFlow op `tf.image.per_image_standardization`.
[5]See `https://gist.github.com/ksimonyan/211839e770f7b538e2d8#description` for the mean RGB values used.

**LSTM** is a two hidden-layer LSTM model (Hochreiter and Schmidhuber, 1997) identical to the model used in Schneider et al. (2019). It uses 128 embedding dimensions and 128 hidden units.

**Transformer** is the "base" model described in (Vaswani et al., 2017). We used it as an autoregressive language model by applying the decoder directly to the sequence of word embeddings for each sentence. Unlike the default implementation, we removed dropout regularization and used separate weight matrices for the input embedding layer and the pre-softmax linear transformation, as we observed these choices led to better performing models.

## C    ESTIMATING TRIAL OUTCOMES VIA BOOTSTRAP

Our tuning protocol corresponds to running trials with quasi-random hyperparameter values sampled uniformly from the search space until $K$ feasible trials are obtained, with $K$ depending on the workload. We then select the best trial, based on our statistic of interest, over those $K$ trials.

We used the following bootstrap procedure to estimate means and uncertainties of our tuning protocol. We ran $N > K$ trials, with $N$ depending on the workload. Then, for each bootstrap sample, we resampled the dataset of $N$ trials with replacement and computed our statistic on the first $K$ trials of the resampled dataset. We collected 100 such bootstrap samples each time, and from those computed the means, $5^{\text{th}}$ percentiles, and $95^{\text{th}}$ percentiles of the bootstrap distribution. We used this procedure to generate the means and error bars for each plot.

Simple CNN on Fashion MNIST used $(K, N) = (100, 500)$; ResNet-32 on CIFAR-100 used $(K, N) = (100, 500)$; ResNet-50 on ImageNet used $(K, N) = (50, 250)$; Transformer on LM1B used $(K, N) = (50, 100)$; VGG on CIFAR-10 with our code used $(K, N) = (50, 250)$ for tuning the learning rate schedule and $(K, N) = (100, 500)$ for tuning the learning rate schedule, $\{\gamma, \beta_1, \beta_2, \epsilon\}$, and $L_2$ regularization; CNN on CIFAR-10 used $(K, N) = (100, 500)$; LSTM on *War and Peace* used $(K, N) = (10, 50)$ for tuning just the learning rate and $(K, N) = (100, 500)$ for tuning the learning rate schedule and $\{\gamma, \beta_1, \beta_2, \epsilon\}$.

The sole exceptions to this bootstrap procedure are the two left panels of Figure 3, for which we used a similar procedure to Wilson et al. (2017) to ensure comparability. For each optimizer, we selected the trial that minimized validation error in our final search space and ran the same hyperparameter values 5 times, reporting the mean, minimum, and maximum test error over those 5 runs in Figure 3. This is slightly different to Wilson et al. (2017), who chose the trial that minimized training error and reported validation error. When tuning the learning rate and $\{\gamma, \epsilon\}$, we used 24 trials per optimizer in the initial search space (which we used to select the final search space), and 16 trials per optimizer in the final search space.

## D    HYPERPARAMETER SEARCH SPACES

When tuning hyperparameters over a large range, we found that our search could sometimes be made more efficient if we parametrized the search space in a way that decorrelated the axes of the space. For example, with MOMENTUM and NESTEROV we observed a clear relationship between the initial learning rate $\eta_0$ and the momentum parameter $\gamma$; smaller values of $\eta_0$ require larger values of $\gamma$ for good performance, and vice versa. Indeed, Shallue et al. (2019) suggested that these optimizers are governed by the "effective learning rate" $\eta_{\text{eff}} = \eta_0/(1 - \gamma)$, and inspired by this, we found that searching over $(\eta_0, \eta_{\text{eff}})$ instead of $(\eta_0, \gamma)$ usually led to a more efficient hyperparameter search. Similarly, with ADAM and NADAM we observed a relationship between the initial learning rate $\alpha_0$ and the $\epsilon$ parameter; larger values of $\alpha_0$ require larger values of $\epsilon$ for good performance, and vice versa. This is not surprising given the analysis in Appendix A that showed that, for large $\epsilon$, $\alpha_0/\epsilon$ is analogous to the effective learning rate of ADAM and NADAM. We found that searching over $(\epsilon, \alpha_0/\epsilon)$ was usually more efficient than searching over $(\epsilon, \alpha_0)$. We used these techniques in a subset of our experiments.

Below we report the search spaces used for our experiments. We include both the initial search spaces used to refine the search spaces, and the final spaces used to generate the plots. When only one search space was used, we denote the initial space as final. $\eta_0$, $\alpha_0$, $1 - \gamma$, $1 - \beta_1$, $1 - \beta_2$, $\epsilon$, and combinations thereof are always tuned on a log scale. The number of samples from each search space is specified in Appendix C.

## D.1 CNN ON FASHION MNIST

We used linear learning rate decay for all experiments. We tuned the number of decay steps within $[0.5, 1.0]$ times the number of training steps and the learning rate decay factor within $\{10^{-3}, 10^{-2}, 10^{-1}\}$. We did not use $L_2$ regularization or weight decay.

| | $\eta_0$ |
|---|---|
| initial | $[10^{-2}, 10^2]$ |
| final | $[10^{-2}, 10^1]$ |

Table 3: SGD

| | $\eta_0$ | $1 - \gamma$ |
|---|---|---|
| initial | $[10^{-4}, 10^2]$ | $[10^{-4}, 1]$ |
| final | $[10^{-5}, 10^1]$ | $[10^{-4}, 1]$ |

Table 4: MOMENTUM

| | $\eta_0$ | $1 - \gamma$ |
|---|---|---|
| initial | $[10^{-4}, 10^2]$ | $[10^{-4}, 1]$ |
| final | $[10^{-4}, 10^1]$ | $[10^{-4}, 1]$ |

Table 5: NESTEROV

| | $\eta_0$ | $1 - \gamma$ | $1 - \rho$ | $\epsilon$ |
|---|---|---|---|---|
| initial | $[10^{-4}, 10^1]$ | $[10^{-2}, 1]$ | $[10^{-4}, 1]$ | $[10^{-5}, 10^1]$ |
| final | $[10^{-5}, 1]$ | $[10^{-2}, 1]$ | $[10^{-3}, 1]$ | $[10^{-10}, 10^{-5}]$ |

Table 6: RMSPROP

| | $\alpha_0$ | $1 - \beta_1$ | $1 - \beta_2$ | $\epsilon$ |
|---|---|---|---|---|
| initial | $[10^{-4}, 10^{-1}]$ | $[10^{-3}, 5 \times 10^{-1}]$ | $[10^{-4}, 10^{-1}]$ | $[10^{-9}, 10^{-5}]$ |
| final | $[10^{-5}, 10^{-1}]$ | $[10^{-3}, 1]$ | $[10^{-4}, 1]$ | $[10^{-10}, 10^{-5}]$ |

Table 7: ADAM

| | $\alpha_0/\epsilon$ | $1 - \beta_1$ | $1 - \beta_2$ | $\epsilon$ |
|---|---|---|---|---|
| initial | $[10^{-2}, 10^4]$ | $[10^{-3}, 1]$ | $[10^{-4}, 1]$ | $[10^{-10}, 10^{10}]$ |
| final | $[10^{-1}, 10^1]$ | $[10^{-3}, 1]$ | $[10^{-4}, 1]$ | $[10^{-6}, 10^{-2}]$ |

Table 8: NADAM

### D.2 RESNET-32 ON CIFAR-10

We used linear learning rate decay for all experiments. We tuned the number of decay steps within $[0.5, 1.0]$ times the number of training steps and the learning rate decay factor $f$ within the values shown in the tables below. $\lambda_{L_2}$ denotes the $L_2$ regularization coefficient.

| | $\eta_0$ | $\lambda_{L_2}$ | $f$ |
|---|---|---|---|
| final | $[10^{-2}, 10^2]$ | $\{10^{-5}, 10^{-4}, 10^{-3}, 10^{-2}\}$ | $\{10^{-4}, 10^{-3}, 10^{-2}, 10^{-1}\}$ |

Table 9: SGD

| | $\eta_0$ | $1 - \gamma$ | $\lambda_{L_2}$ | $f$ |
|---|---|---|---|---|
| final | $[10^{-4}, 10^2]$ | $[10^{-3}, 1]$ | $\{10^{-5}, 10^{-4}, 10^{-3}, 10^{-2}\}$ | $\{10^{-4}, 10^{-3}, 10^{-2}, 10^{-1}\}$ |

Table 10: MOMENTUM

| | $\eta_0$ | $1 - \gamma$ | $\lambda_{L_2}$ | $f$ |
|---|---|---|---|---|
| initial | $[10^{-4}, 10^2]$ | $[10^{-4}, 10^1]$ | $10^{-4}$ | $\{10^{-3}, 10^{-2}, 10^{-1}\}$ |
| final | $[10^{-4}, 10^1]$ | $[10^{-4}, 1]$ | $\{10^{-5}, 10^{-4}, 10^{-3}, 10^{-2}\}$ | $\{10^{-4}, 10^{-3}, 10^{-2}, 10^{-1}\}$ |

Table 11: NESTEROV

| | $\eta_0$ | $1 - \gamma$ | $1 - \rho$ | $\epsilon$ | $\lambda_{L_2}$ | $f$ |
|---|---|---|---|---|---|---|
| initial | $[10^{-4}, 10^1]$ | $[10^{-2}, 1]$ | $[10^{-4}, 1]$ | $[10^{-5}, 10^1]$ | $10^{-4}$ | $\{10^{-3}, 10^{-2}, 10^{-1}\}$ |
| final | $[10^{-4}, 10^1]$ | $[10^{-3}, 1]$ | $[10^{-4}, 1]$ | $[10^{-5}, 10^1]$ | $\{10^{-5}, 10^{-4} \ 10^{-3}, 10^{-2}\}$ | $\{10^{-4}, 10^{-3} \ 10^{-2}, 10^{-1}\}$ |

Table 12: RMSPROP

| | $\alpha_0$ | $1 - \beta_1$ | $1 - \beta_2$ | $\epsilon$ | $\lambda_{L_2}$ | $f$ |
|---|---|---|---|---|---|---|
| initial | $[10^{-4}, 10^{-1}]$ | $[10^{-3}, 5 \times 10^{-1}]$ | $[10^{-4}, 10^{-1}]$ | $[10^{-9}, 10^{-5}]$ | $10^{-4}$ | $\{10^{-3}, 10^{-2} \ 10^{-1}\}$ |
| final | $[10^{-3}, 10^1]$ | $[10^{-3}, 1]$ | $[10^{-4}, 10^{-1}]$ | $[10^{-5}, 10^1]$ | $\{10^{-5}, 10^{-4} \ 10^{-3}, 10^{-2}\}$ | $\{10^{-4}, 10^{-3} \ 10^{-2}, 10^{-1}\}$ |

Table 13: ADAM

| | $\alpha_0 / \epsilon$ | $1 - \beta_1$ | $1 - \beta_2$ | $\epsilon$ | $\lambda_{L_2}$ | $f$ |
|---|---|---|---|---|---|---|
| initial | $[10^{-2}, 10^4]$ | $[10^{-3}, 1]$ | $[10^{-4}, 1]$ | $[10^{-10}, 10^{10}]$ | $\{10^{-5}, 10^{-4} \ 10^{-3}, 10^{-2}\}$ | $\{10^{-4}, 10^{-3} \ 10^{-2}, 10^{-1}\}$ |
| final | $[10^{-2}, 1]$ | $[10^{-3}, 1]$ | $[10^{-4}, 1]$ | $[1, 10^4]$ | $\{10^{-5}, 10^{-4} \ 10^{-3}, 10^{-2}\}$ | $\{10^{-4}, 10^{-3} \ 10^{-2}, 10^{-1}\}$ |

Table 14: NADAM

### D.3 RESNET-50 ON IMAGENET

We used linear learning rate decay for all experiments. We tuned the number of decay steps within $[0.5, 1.0]$ times the number of training steps and the learning rate decay factor $f$ within the values shown in the tables below. $\lambda_{\text{wd}}$ denotes the weight decay coefficient and $\tau$ denotes the label smoothing coefficient.

|  | $\eta_0$ | $\lambda_{\text{wd}}$ | $\tau$ | $f$ |
|---|---|---|---|---|
| initial | $[10^{-2}, 10^{1}]$ | $[10^{-5}, 10^{-2}]$ | $\{0, 10^{-2}, 10^{-1}\}$ | $\{10^{-4}, 10^{-3}, 10^{-2}, 10^{-1}\}$ |
| final | $[1, 10^{2}]$ | $[10^{-4}, 10^{-3}]$ | $10^{-1}$ | $\{10^{-4}, 10^{-3}, 10^{-2}, 10^{-1}\}$ |

Table 15: SGD

|  | $\eta_0$ | $1 - \gamma$ | $\lambda_{\text{wd}}$ | $\tau$ | $f$ |
|---|---|---|---|---|---|
| initial | $[10^{-3}, 1]$ | $[10^{-3}, 1]$ | $[10^{-5}, 10^{-2}]$ | $\{0, 10^{-2}, 10^{-1}\}$ | $\{10^{-4}, 10^{-3}, 10^{-2}, 10^{-1}\}$ |
| final | $[10^{-2}, 1]$ | $[10^{-2}, 1]$ | $[10^{-4}, 10^{-3}]$ | $10^{-2}$ | $\{10^{-4}, 10^{-3}, 10^{-2}, 10^{-1}\}$ |

Table 16: MOMENTUM

|  | $\eta_0$ | $1 - \gamma$ | $\lambda_{\text{wd}}$ | $\tau$ | $f$ |
|---|---|---|---|---|---|
| initial | $[10^{-3}, 1]$ | $[10^{-3}, 1]$ | $[10^{-5}, 10^{-2}]$ | $\{0, 10^{-2}, 10^{-1}\}$ | $10^{-3}$ |
| final | $[10^{-2}, 1]$ | $[10^{-3}, 1]$ | $[10^{-4}, 10^{-3}]$ | $0$ | $\{10^{-4}, 10^{-3}, 10^{-2}, 10^{-1}\}$ |

Table 17: NESTEROV

|  | $\eta_0/\sqrt{\epsilon}$ | $1 - \gamma$ | $1 - \rho$ | $\epsilon$ | $\lambda_{\text{wd}}$ | $\tau$ | $f$ |
|---|---|---|---|---|---|---|---|
| initial | $[10^{-2}, 10^{4}]$ | $0.1$ | $[10^{-4}, 1]$ | $[10^{-10}, 10^{10}]$ | $[10^{-5}, 10^{-2}]$ | $\{0, 10^{-2}, 10^{-1}\}$ | $10^{-3}$ |
| final | $[10^{-2}, 1]$ | $0.1$ | $[10^{-2}, 1]$ | $[10^{-8}, 10^{-3}]$ | $[10^{-4}, 10^{-3}]$ | $0$ | $\{10^{-4}, 10^{-3} \\ 10^{-2}, 10^{-1}\}$ |

Table 18: RMSPROP

|  | $\alpha_0/\epsilon$ | $1 - \beta_1$ | $\epsilon$ | $\lambda_{\text{wd}}$ | $\tau$ | $f$ |
|---|---|---|---|---|---|---|
| initial | $[1, 10^{2}]$ | $[10^{-3}, 1]$ | $[1, 10^{4}]$ | $[10^{-5}, 10^{-3}]$ | $\{0, 10^{-2}, 10^{-1}\}$ | $10^{-3}$ |
| final | $[1, 10^{2}]$ | $[10^{-2}, 1]$ | $[10^{-2}, 10^{2}]$ | $10^{-4}$ | $10^{-1}$ | $\{10^{-4}, 10^{-3} \\ 10^{-2}, 10^{-1}\}$ |

Table 19: ADAM

|  | $\alpha_0/\epsilon$ | $1 - \beta_1$ | $\epsilon$ | $\lambda_{\text{wd}}$ | $\tau$ | $f$ |
|---|---|---|---|---|---|---|
| initial | $[10^{-1}, 10^{3}]$ | $[10^{-3}, 1]$ | $[10^{-2}, 10^{10}]$ | $[10^{-5}, 10^{-2}]$ | $\{0, 10^{-2}, 10^{-1}\}$ | $10^{-3}$ |
| final | $[1, 10^{2}]$ | $[10^{-3}, 1]$ | $[10^{3}, 10^{7}]$ | $10^{-4}$ | $10^{-1}$ | $10^{-3}$ |

Table 20: NADAM

### D.4 Transformer on LM1B

We used linear learning rate decay for all experiments. We tuned the number of decay steps within $[0.5, 1.0]$ times the number of training steps and the learning rate decay factor within $\{10^{-4}, 10^{-3}, 10^{-2}, 10^{-1}, 1\}$.

|  | $\eta_0$ |
|---|---|
| final | $[10^{-4}, 10^{-1}]$ |

Table 21: SGD

|  | $\eta_0$ | $1 - \gamma$ |
|---|---|---|
| final | $[10^{-4}, 10^{-1}]$ | $[10^{-4}, 1]$ |

Table 22: Momentum

|  | $\eta_0$ | $1 - \gamma$ |
|---|---|---|
| final | $[10^{-4}, 10^{-1}]$ | $[10^{-4}, 1]$ |

Table 23: Nesterov

|  | $\eta_0$ | $1 - \gamma$ | $1 - \rho$ | $\epsilon$ |
|---|---|---|---|---|
| initial | $[10^{-4}, 10^1]$ | $[10^{-2}, 1]$ | $[10^{-2}, 1]$ | $[10^{-12}, 10^{10}]$ |
| final | $[10^{-6}, 10^{-2}]$ | $[10^{-2}, 1]$ | $[10^{-3}, 1]$ | $[10^{-7}, 10^{-1}]$ |

Table 24: RMSProp

|  | $\alpha_0$ | $1 - \beta_1$ | $1 - \beta_2$ | $\epsilon$ |
|---|---|---|---|---|
| initial | $[10^{-5}, 10^{-2}]$ | $[10^{-3}, 5 \times 10^{-1}]$ | $[10^{-4}, 10^{-1}]$ | $[10^{-9}, 10^{-5}]$ |
| final | $[10^{-4}, 10^{-2}]$ | $[10^{-3}, 1]$ | $[10^{-5}, 10^{-1}]$ | $[10^{-7}, 10^{-2}]$ |

Table 25: Adam

|  | $\alpha_0$ | $1 - \beta_1$ | $1 - \beta_2$ | $\epsilon$ |
|---|---|---|---|---|
| final | $[10^{-5}, 10^{-2}]$ | $[10^{-3}, 1]$ | $[10^{-5}, 10^{-1}]$ | $[10^{-9}, 10^{-5}]$ |

Table 26: NAdam

### D.5 VGG ON CIFAR-10 USING CODE FROM WILSON ET AL. (2017)

#### D.5.1 GRID SEARCH OVER LEARNING RATE

We tuned over the same grid of initial learning rate values for each optimizer as Wilson et al. (2017). As in Wilson et al. (2017), we decayed the initial learning rate by a factor of $0.5$ every 25 epochs and used a fixed $L_2$ regularization coefficient of $0.0005$.

#### D.5.2 TUNING LEARNING RATE & $\{\gamma, \epsilon\}$

We used our quasi-random tuning protocol to tune over the initial learning rate, MOMENTUM's $\gamma$, RMSPROP's $\epsilon$, and ADAM's $\epsilon$. As in Wilson et al. (2017), we decayed the initial learning rate by a factor of $0.5$ every 25 epochs and used a fixed $L_2$ regularization coefficient of $0.0005$.

|  | $\eta_0$ |
|---|---|
| initial | $[10^{-3}, 1]$ |
| final | $[10^{-1}, 10^1]$ |

Table 27: SGD

|  | $\eta_0$ | $1 - \gamma$ |
|---|---|---|
| initial | $[10^{-3}, 1]$ | $[10^{-3}, 1]$ |
| final | $[10^{-1}, 10^1]$ | $[10^{-1}, 1]$ |

Table 28: MOMENTUM

|  | $\epsilon$ | $\alpha_0/\sqrt{\epsilon}$ |
|---|---|---|
| initial | $[10^{-10}, 10^{10}]$ | $[10^{-2}, 10^4]$ |
| final | $[10^{-2}, 10^2]$ | $[10^{-1}, 10^1]$ |

Table 29: RMSPROP

|  | $\epsilon$ | $\alpha_0/\epsilon$ |
|---|---|---|
| initial | $[10^{-10}, 10^{10}]$ | $[10^{-2}, 10^4]$ |
| final | $[10^6, 10^{10}]$ | $[10^{-1}, 10^1]$ |

Table 30: ADAM

## D.6 VGG ON CIFAR-10 USING OUR CODE

We used linear learning rate decay for all experiments. We tuned the number of decay steps within $[0.5, 1.0]$ times the number of training steps and the learning rate decay factor within $\{10^{-4}, 10^{-3}, 10^{-2}, 10^{-1}\}$.

### D.6.1 TUNING LEARNING RATE SCHEDULE

We fixed all optimizer hyperparameters excluding the learning rate to match those specified in Wilson et al. (2017). As in Wilson et al. (2017), we used a fixed $L_2$ regularization coefficient of 0.0005.

|  | $\eta_0$ (SGD) | $\eta_0$ (MOMENTUM) | $\eta_0$ (RMSPROP) | $\alpha_0$ (ADAM) |
|---|---|---|---|---|
| initial | $[10^{-3}, 10^1]$ | $[10^{-3}, 10^1]$ | $[10^{-5}, 10^{-1}]$ | $[10^{-5}, 10^{-1}]$ |
| final | 1.0 | $[10^{-2}, 1]$ | $[10^{-4}, 10^{-2}]$ | $[10^{-5}, 10^{-1}]$ |

Table 31: Learning rate search ranges.

### D.6.2 TUNING LEARNING RATE SCHEDULE & $\{\gamma, \beta_1, \beta_2, \epsilon, \lambda_{L_2}\}$

|  | $\eta_0$ | $\lambda_{L_2}$ |
|---|---|---|
| initial | $[10^{-3}, 10^1]$ | $[10^{-5}, 10^{-2}]$ |
| final | $[10^{-2}, 1]$ | $[10^{-3}, 10^{-1}]$ |

Table 32: SGD

|  | $\eta_0$ | $1 - \gamma$ | $\lambda_{L_2}$ |
|---|---|---|---|
| initial | $[10^{-3}, 10^1]$ | $[10^{-3}, 1]$ | $[10^{-5}, 10^{-2}]$ |
| final | $[10^{-2}, 1]$ | $[10^{-1}, 1]$ | $[10^{-3}, 10^{-1}]$ |

Table 33: MOMENTUM

|  | $\alpha_0/\sqrt{\epsilon}$ | $1 - \gamma$ | $1 - \rho$ | $\epsilon$ | $\lambda_{L_2}$ |
|---|---|---|---|---|---|
| initial | $[10^{-2}, 10^4]$ | $[10^{-3}, 1]$ | $[10^{-3}, 1]$ | $[10^{-10}, 10^{10}]$ | $[10^{-5}, 10^{-2}]$ |
| final | $[10^{-2}, 1]$ | $[10^{-1}, 1]$ | $[10^{-3}, 10^{-2}]$ | $[10^2, 10^6]$ | $[10^{-3}, 10^{-1}]$ |

Table 34: RMSPROP

|  | $\alpha_0/\epsilon$ | $1 - \beta_1$ | $1 - \beta_2$ | $\epsilon$ | $\lambda_{L_2}$ |
|---|---|---|---|---|---|
| initial | $[10^{-2}, 10^4]$ | $[10^{-3}, 1]$ | $[10^{-4}, 10^{-1}]$ | $[10^{-10}, 10^{10}]$ | $[10^{-5}, 10^{-2}]$ |
| final | $[10^{-2}, 10^1]$ | $[10^{-1}, 1]$ | $[10^{-4}, 10^{-1}]$ | $[10^6, 10^{10}]$ | $[10^{-3}, 10^{-1}]$ |

Table 35: ADAM

### D.7 CNN ON CIFAR-100

#### D.7.1 TUNING CONSTANT LEARNING RATE

We fixed all optimizer hyperparameters excluding the learning rate to match those specified in Schneider et al. (2019).

|  | $\eta$ (SGD) | $\eta$ (MOMENTUM) | $\alpha$ (ADAM) |
|---|---|---|---|
| initial | $[10^{-2}, 1]$ | $[10^{-4}, 1]$ | $[10^{-5}, 10^{-2}]$ |
| final | $[10^{-1}, 1]$ | $[10^{-3}, 10^{-2}]$ | $[10^{-4}, 10^{-3}]$ |

Table 36: Learning rate search ranges.

#### D.7.2 TUNING LEARNING RATE SCHEDULE

We used linear learning rate decay, and tuned the number of decay steps within $[0.5, 1.0]$ times the number of training steps and the learning rate decay factor within $\{10^{-4}, 10^{-3}, 10^{-2}, 10^{-1}\}$.

|  | $\eta_0$ (SGD) | $\eta_0$ (MOMENTUM) | $\alpha_0$ (ADAM) |
|---|---|---|---|
| initial | $[10^{-2}, 1]$ | $[10^{-4}, 1]$ | $[10^{-5}, 10^{-2}]$ |
| final | $[10^{-1}, 1]$ | $[10^{-3}, 10^{-1}]$ | $[10^{-4}, 10^{-3}]$ |

Table 37: Learning rate search ranges.

#### D.7.3 TUNING CONSTANT LEARNING RATE & $\{\gamma, \beta_1, \beta_2, \epsilon\}$

For SGD, we reused the results from Appendix D.7.1, since there were no additional hyperparameters to tune.

|  | $\eta$ | $1 - \gamma$ |
|---|---|---|
| final | $[10^{-4}, 1]$ | $[10^{-3}, 1]$ |

Table 38: MOMENTUM

|  | $\alpha/\epsilon$ | $1 - \beta_1$ | $1 - \beta_2$ | $\epsilon$ |
|---|---|---|---|---|
| initial | $[10^{-2}, 10^{-2}]$ | $[10^{-3}, 1]$ | $[10^{-4}, 10^{-1}]$ | $[10^{-10}, 10^{-10}]$ |
| final | $[10^{-1}, 10^{-1}]$ | $[10^{-2}, 1]$ | $[10^{-4}, 10^{-1}]$ | $[10^2, 10^6]$ |

Table 39: ADAM

#### D.7.4 TUNING LEARNING RATE SCHEDULE & $\{\gamma, \beta_1, \beta_2, \epsilon\}$

We used linear learning rate decay, and tuned the number of decay steps within $[0.5, 1.0]$ times the number of training steps and the learning rate decay factor within $\{10^{-4}, 10^{-3}, 10^{-2}, 10^{-1}\}$. For SGD, we reused the results from Appendix D.7.2, since there were no additional hyperparameters to tune.

|  | $\eta_0$ | $1 - \gamma$ |
|---|---|---|
| initial | $[10^{-4}, 1]$ | $[10^{-2}, 1]$ |
| final | $[10^{-3}, 10^{-1}]$ | $[10^{-3}, 10^{-1}]$ |

Table 40: MOMENTUM

|  | $\alpha_0/\epsilon$ | $1 - \beta_1$ | $1 - \beta_2$ | $\epsilon$ |
|---|---|---|---|---|
| initial | $[10^{-1}, 10^1]$ | $[10^{-3}, 1]$ | $[10^{-4}, 10^{-1}]$ | $[10^2, 10^6]$ |
| final | $[10^{-1}, 10^1]$ | $[10^{-3}, 1]$ | $[10^{-5}, 10^{-2}]$ | $[10^2, 10^6]$ |

Table 41: ADAM

## D.8  LSTM ON WAR AND PEACE

### D.8.1  TUNING CONSTANT LEARNING RATE

|  | $\eta$ |
|---|---|
| final | $[10^{-2}, 10^1]$ |

Table 42: SGD

|  | $\eta$ | $1 - \gamma$ |
|---|---|---|
| final | $[10^{-4}, 1]$ | 0.99 |

Table 43: MOMENTUM

|  | $\alpha/\epsilon$ | $1 - \beta_1$ | $1 - \beta_2$ | $\epsilon$ |
|---|---|---|---|---|
| final | $[10^{-5}, 10^{-2}]$ | 0.9 | 0.999 | $10^{-8}$ |

Table 44: ADAM

### D.8.2  TUNING LEARNING RATE SCHEDULE & $\{\gamma, \beta_1, \beta_2, \epsilon\}$

We used linear learning rate decay, and tuned the number of decay steps within $[0.5, 1.0]$ times the number of training steps and the learning rate decay factor $f$ within the values shown in the tables below.

|  | $\eta_0$ | $f$ |
|---|---|---|
| initial | $[10^{-3}, 10^1]$ | $\{10^{-4}, 10^{-3}, 10^{-2}, 10^{-1}\}$ |
| final | $[1, 10^1]$ | $\{10^{-4}, 10^{-3}, 10^{-2}, 10^{-1}\}$ |

Table 45: SGD

|  | $\eta_0$ | $1 - \gamma$ | $f$ |
|---|---|---|---|
| initial | $[10^{-4}, 1]$ | $[10^{-3}, 1]$ | $\{10^{-4}, 10^{-3}, 10^{-2}, 10^{-1}\}$ |
| final | $[10^{-1}, 10^1]$ | $[10^{-2}, 1]$ | $\{10^{-4}, 10^{-3}, 10^{-2}, 10^{-1}\}$ |

Table 46: MOMENTUM

|  | $\alpha_0/\epsilon$ | $1 - \beta_1$ | $1 - \beta_2$ | $\epsilon$ | $f$ |
|---|---|---|---|---|---|
| initial | $[10^{-2}, 10^4]$ | $[10^{-3}, 1]$ | $[10^{-4}, 10^{-1}]$ | $[10^{-10}, 10^{10}]$ | $\{10^{-4}, 10^{-3}, 10^{-2}, 10^{-1}\}$ |
| final | $[1, 10^2]$ | $[10^{-2}, 1]$ | 0.999 | $[1, 10^4]$ | $10^{-3}$ |

Table 47: ADAM

# E    ADDITIONAL PLOTS

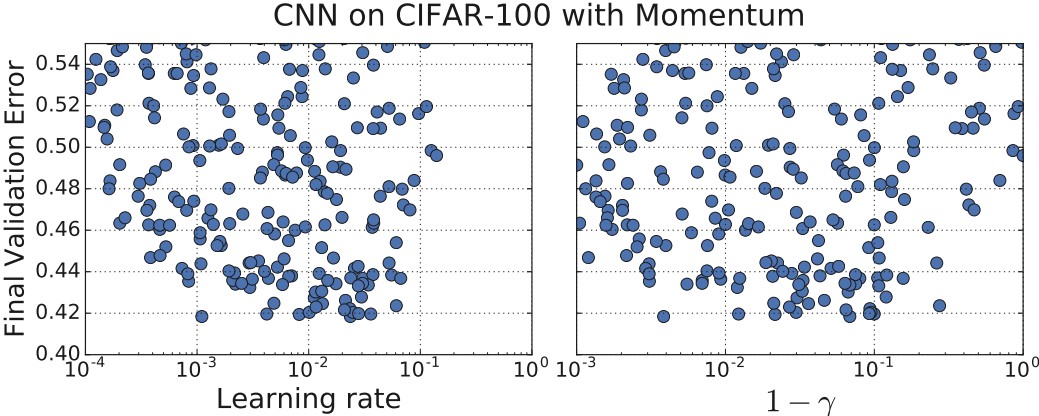

Figure 5: Example plot of final validation error projected onto the axes of the hyperparameter space. We consider this search space to be appropriate because the optimal values are away from the search space boundaries.

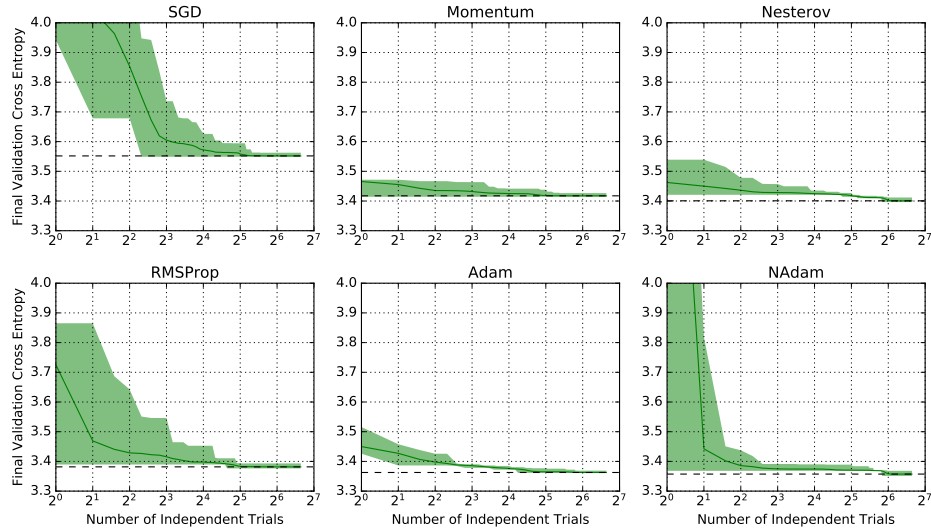

Figure 6: Validation performance of the best trial mostly converges with as few as $2^4$ hyperparameter tuning trials for Transformer on LM1B. Shaded regions indicate $5^{th}$ and $95^{th}$ percentiles estimated with bootstrap sampling (see Appendix C). The search spaces can be found in Appendix D.4.

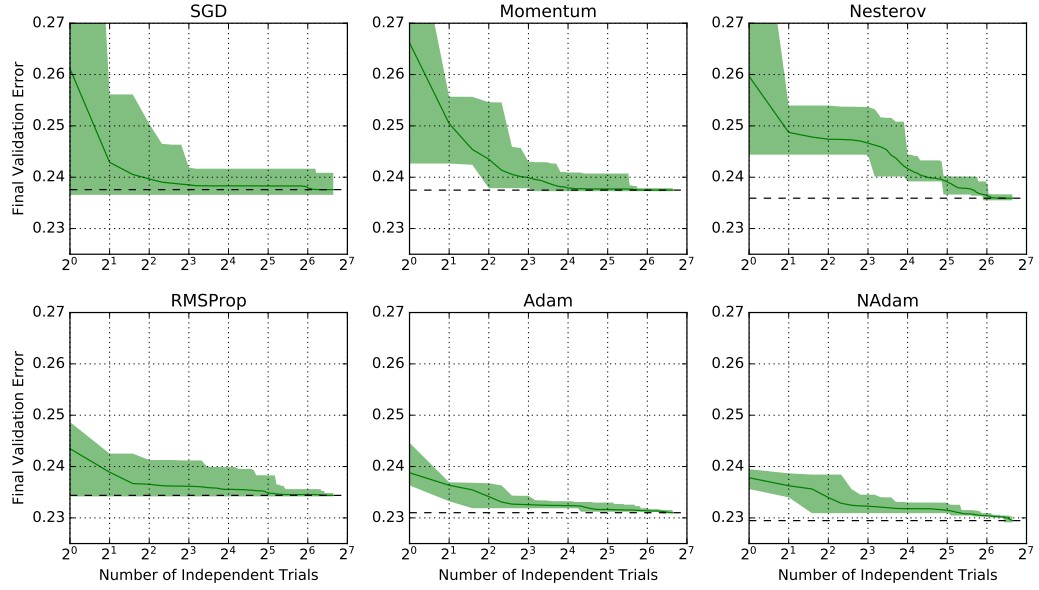

Figure 7: Validation performance of the best trial mostly converges with as few as $2^4$ hyperparameter tuning trials for ResNet-50 in ImageNet. Shaded regions indicate $5^{th}$ and $95^{th}$ percentile estimated with bootstrap sampling (see Appendix C). The search spaces can be found in D.3.

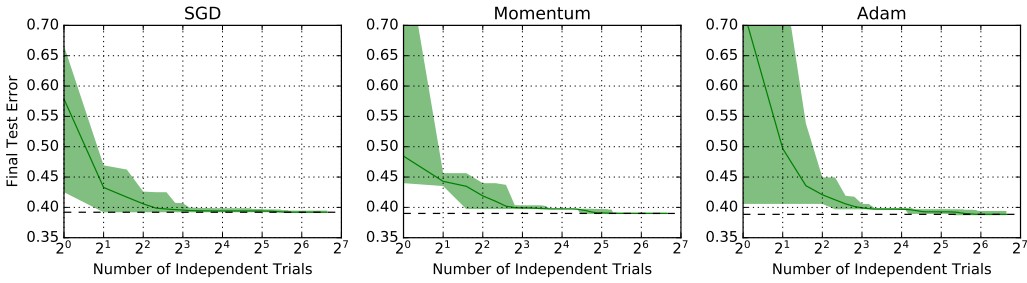

Figure 8: Test performance of the best trial mostly converges with as few as $2^3$ hyperparameter tuning trials for a 2-layer LSTM on *War and Peace*. Shaded regions indicate $5^{th}$ and $95^{th}$ percentile estimated with bootstrap sampling (see Appendix C). The search spaces can be found in D.8.2.

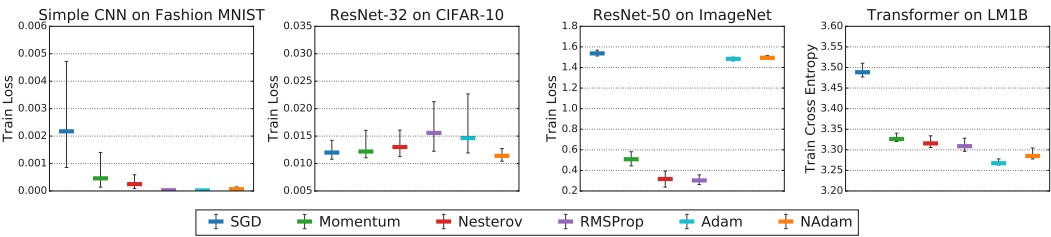

Figure 9: The relative performance of optimizers is consistent with the inclusion relationships when we select for lowest training loss. Note that SGD, ADAM, and NADAM for ResNet-50 on ImageNet used label smoothing in their final search spaces (see Section D.3), which makes their loss values incommensurate with the other optimizers. This is because their final search spaces were optimized to minimize validation error—if we had optimized their search spaces to minimize training error instead, we would not have used label smoothing, and we expect their training loss values would be consistent with the inclusion relationships.

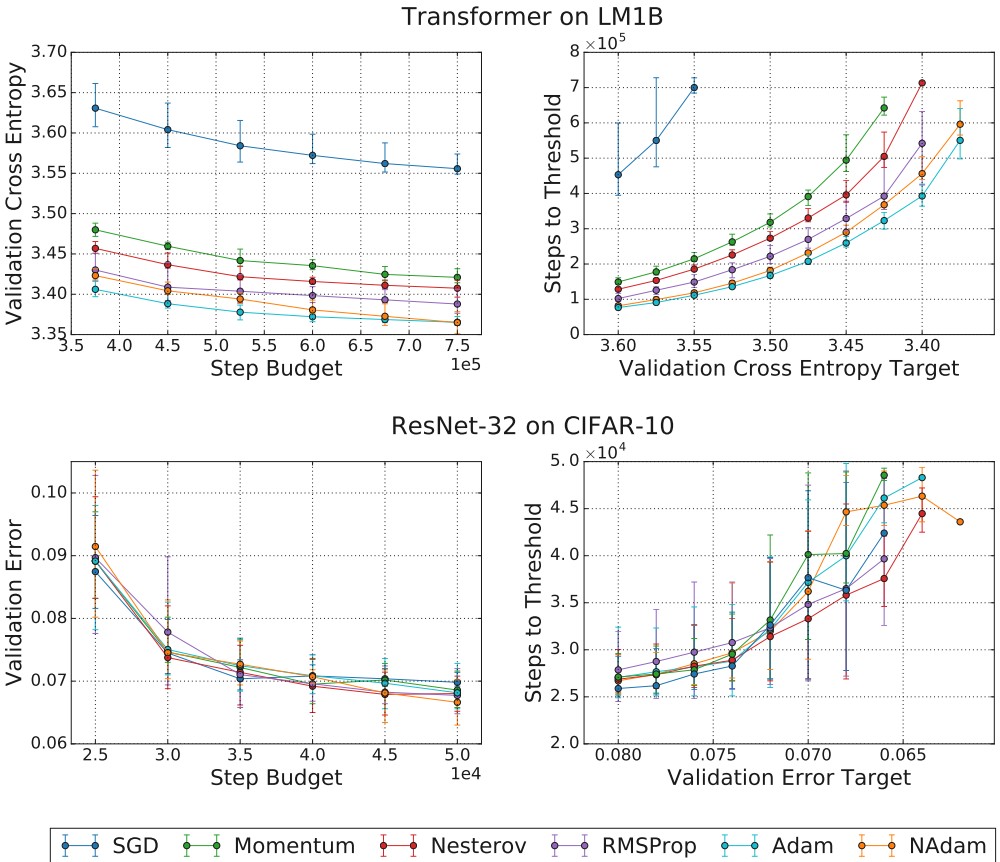

Figure 10: Our results confirming the relevance of optimizer inclusion relationships do not depend on the exact step budgets or error targets we chose.

