# OpenReview forum: "On Empirical Comparisons of Optimizers for Deep Learning"
_ICLR.cc/2020/Conference — Reject_

### Official Review · AnonReviewer2 · 2019-10-21
**Official Blind Review #2**

**Rating:** 6

**Review:**

This paper presents experimental data supporting the claim that the under aggressive hyper-parameter tuning different optimizers are essentially ranked by inclusion --- if the hyper-parameters of method A can simulate any setting of the hyper-parameters of method B then under aggressive hyper-parameter tuning A will dominate B.  One way to achieve this rather trivially is to do a hyper-parameter search for B and then set the hyper-parameters of A so that A is simulating B.  But the point here is that direct and feasible tuning of A with dominate B even in the case where A has more hyper-parameters and where hyper-parameter optimization of A would seem to be more difficult.  An important conclusion is that without loss of generality one can always use Adam even in vision where SGD is currently the dominant optimizer used in practice.  Another important conclusion is that quasi-random hyper-parameter optimization is quite effective.
I find the claims to be intuitively plausible and I find the empirical results compelling.

Just a couple minor complaints.  First,"Hyper-parameter" please not "meta-parameter".  I find the attempt to overturn standard usage inappropriate.  Second, I found most of section 3 uninformative.  I don't think algorithm 1, or the definition of a first order optimizer adds anything to the paper.  Inclusion can be easily defined in complete generality for any parameterized algorithm.

**Experience Assessment:**

I have read many papers in this area.

**Review Assessment: Checking Correctness Of Derivations And Theory:**

I assessed the sensibility of the derivations and theory.

**Review Assessment: Checking Correctness Of Experiments:**

I assessed the sensibility of the experiments.

**Review Assessment: Thoroughness In Paper Reading:**

I made a quick assessment of this paper.

---

> ### Author Response · Authors · 2019-11-09
> **We have incorporated your feedback into a revised version of the manuscript**
>
> We thank the reviewer for their encouraging feedback. We incorporated this feedback into a new revision, which we believe is much stronger.
>
> We agree with both Reviewers 2 & 3 that Section 3 could have been more concise and informative. We trimmed and restructured this section, removed Algorithm 1, and moved the update rule definitions into the main body.
>
> Given that at least 2/3 of the reviewers find the term "hyperparameter" clearer than "metaparameter" we have adopted that language in the latest revision of our paper.
>
> We hope that we have adequately addressed all the concerns in this review and that the reviewer will consider raising their score accordingly. Please let us know if you believe additional issues remain with the newest version.

---

### Official Review · AnonReviewer1 · 2019-10-23
**Official Blind Review #1**

**Rating:** 1

**Review:**



First, I would like to note that the claim that SGD with momentum is a special case of Adam with large epsilon is technically wrong because Adam also includes the bias-corrected momentum estimates which SGD with momentum does not consider. It might seem like a small difference, however it is a form of learning rate schedule which most users of Adam are not aware of. In practice, however, Adam with large epsilon can approximate SGD with momentum. Just don't claim the equivalent since it is not there.

I have some difficulties understanding the contribution of the paper. For example
"When tuning all available metaparameters under a realistic protocol at scales common in deep learning,
we find that more general update rules never underperform their special cases."
In practice you do adjust hyperparameter search spaces to fit your conclusions, e.g., "We found that searching over (epsilon, alpha0/epsilon) was more efficient than searching over (epsilon, alpha)." Again, this alone invalidates your experimental setup since you biased it in order to fit your conclusion: "was more efficient" was found after running some prior experiments.
Another situation where your experimental setup is unfairly tuned is when you used different hyperparameter ranges for similar hyperparameter, e.g. see D.2 for ResNet-32 on CIFAR-10 where 6 orders of magnitute difference was used for the initial learning of Momentum and 3 orders of magnitude difference for the initial learning rate of Adam. Similarly, there is a difference of 10x for ImageNet experiments.

The paper suggests that 16 experiments is enough to produce good results. First, one should not forget the  special arrangements (see above) done for hyperparameter search space. Second, for any person working in black-box optimization it is clear that 16 experiments is next to nothing. It should give you something good in 1D, possibly in 2D if your search range is narrow. This is absolutely nothing in larger dimensions (providing that your benchmark in not super trivial and your hyperparameter search space is not absolutely boring when you already narrowed it around the optimum). After 16 evaluations you get pretty bad settings for most algorithms.

Update:

The paper uses a naive hyperparameter optimizer and runs it for a very small budget. The latter likely affects the conclusion of the paper that different training algorithms perform similarly. The authors seem to accept it by mentioning that this is the case for their tuning protocol/budget.

If we would like to compare different training algorithms, we should optimize them on a set of problems using 2-3 state-of-the-art hyperparameter optimizers. Then, we should study how the best seen solutions so far and their robustness  change as a function of the computation budget (the maximum budget should be large enough). Then, one would see that the results are not that different for small budgets (a boring result) and somewhat different for larger budgets. Showing only the boring part seems more misleading than useful.

Update#2:
As I mentioned in my review, Adam with large epsilon is not equivalent to momentum SGD but only approximates the latter. This is because the original Adam has a bias correction term and even if the same *global* learning rate schedule is used both for Adam with large epsilon and momentum SGD, they are not equivalent. In order to obtain the exact equivalence, one would need to either
1) drop the bias correction term of Adam and thus modify the algorithm in order to satisfy the claimed equivalence
or
2) set a particular learning rate *for each batch pass* of Adam to simulate the effect of the bias correction term, this leads to a large number of hyperparameters - as many as the number of batch passes, this is intractable (the setup of the authors does not optimize such batch-wise hyperparameters, they are defined by a global scheduler as a function of batch/epoch index).
If you avoid these modifications, then you can't claim the equivalence but only an approximation. If you don't have the equivalence of the two approaches and so momentum SGD is not a particular case of Adam, then the following sentence from the abstract is false: " As tuning effort grows without bound, more general optimizers should never underperform the ones they can approximate (i.e., Adam should never perform worse than momentum)". Again, strictly speaking, it is false that "Adam should never perform worse than momentum" because momentum SGD is not a particular case of the original Adam *unless* you drop the bias correction term or simulate it with tons of hyperparameters, one learning rate value per batch pass. Any global learning rate schedule *used for both* algorithms will not solve the issue because the bias correction term will remain. If you don't modify the learning rate schedule of Adam but only of momentum SGD, then you basically adjust your SGD by moving some part of Adam in it to claim the equivalence of the two, such actions can make pretty much every second algorithm equivalent to another.

My main concern is described in the first Update. It is trivial that a more general optimizer is capable to perform at least as good as its particular case. What is not trivial is to clarify the interplay of computational budgets spent on hyperparameter tuning vs number of hyperparameters vs performance over time.

**Experience Assessment:**

I have published in this field for several years.

**Review Assessment: Checking Correctness Of Derivations And Theory:**

I carefully checked the derivations and theory.

**Review Assessment: Checking Correctness Of Experiments:**

I assessed the sensibility of the experiments.

**Review Assessment: Thoroughness In Paper Reading:**

I read the paper thoroughly.

---

> ### Author Response · Authors · 2019-11-09
> **Insufficiency of 16 trials**
>
> Thanks for pointing out the overly strong language about the efficiency of the final search spaces we found. We have revised the manuscript to avoid these overly strong claims and, since it is not a crucial part of our argument, moved the figure in question to the appendix.

---

> > ### Comment · AnonReviewer1 · 2019-11-14
> > **Re**
> >
> > The language used is not the main problem. The main problem is that the results and conclusions are  affected by this choice of a small budget and bad hyperparameter optimizer.

---

> > > ### Author Response · Authors · 2019-11-15
> > > **Our budget was not small compared to previous work and test error shows we tuned well**
> > >
> > > We ran 100 trials on most workloads with the exception of 50 for both ResNet-50 on ImageNet and Transformer on LM1B (we repeated all these experiments multiple times to get error bars). This is more trials than standard practice for hyperparameter tuning on these workloads.
> > >
> > > We get better test error than the optimizer comparisons of Wilson et al. 2017, Schneider et al. 2019, and our ImageNet results are better than Goyal et al., 2017.
> > >
> > > Can you provide a reference that compares optimizers on any of our workloads and uses dramatically more tuning trials?

---

> > > > ### Comment · AnonReviewer1 · 2019-11-15
> > > > **Re**
> > > >
> > > > From ICLR 2016: https://arxiv.org/pdf/1604.07269.pdf

---

> ### Author Response · Authors · 2019-11-09
> **Technicalities in approximating SGD with momentum using Adam**
>
> Our paper is correct in stating that SGD with momentum is a special case of Adam in the limit of large epsilon because alpha is allowed to depend on the step number. We have added an extra note in the paper to make this clearer. Adam’s bias correction does indeed act as a form of learning rate schedule (as the review pointed out), but we can always choose the schedule for Adam’s alpha to exactly match an arbitrary learning rate schedule of SGD with momentum. This schedule is listed explicitly in Appendix A.
>
> Of course, practitioners cannot tune Adam over all possible learning rate schedules nor over arbitrarily large values of epsilon, so it is not guaranteed that a practitioner will actually find the hyperparameter values that allow Adam to match or exceed the performance of SGD with momentum. This is the point of our paper! We demonstrate that even if Adam is tuned over finite values of epsilon and linear learning rate schedules, it always matches or exceeds the performance of SGD with momentum in the workloads we considered.

---

> > ### Comment · AnonReviewer1 · 2019-11-14
> > **Re**
> >
> > >> Our paper is correct in stating that SGD with momentum is a special case of Adam in the limit of large epsilon because alpha is allowed to depend on the step number.
> >
> > It is not correct because you can't go from one to another without changing another parts of the algorithm,  here, the bias correction or learning rate schedule.

---

> > > ### Author Response · Authors · 2019-11-15
> > > **Re**
> > >
> > > We provide a proof in Appendix A that Adam includes Momentum under the definition of optimizer inclusions in Section 3. If you are saying the proof is wrong, please provide a counterexample or point out a mistake in the proof so we can understand your point.
> > >
> > > Even if one considers the same optimizer with a different learning rate schedule to be a different optimizer, our proof still correctly shows that for an arbitrary schedule A, there exists a schedule B such that Adam-with-schedule-B is equivalent to Momentum-with-schedule-A. But we do not believe it is useful to consider the learning rate schedule as “part of the algorithm” since, in practice, people use all kinds of different schedules with Adam (or any of the other popular algorithms).

---

> > > > ### Comment · AnonReviewer1 · 2019-11-15
> > > > **Re**
> > > >
> > > > >> We provide a proof in Appendix A that Adam includes Momentum under the definition of optimizer inclusions in Section 3. If you are saying the proof is wrong, please provide a counterexample or point out a mistake in the proof so we can understand your point.
> > > >
> > > > It does not deal with Adam but with YourAdam where you drop the bias correction term.
> > > >
> > > > >> Even if one considers the same optimizer with a different learning rate schedule to be a different optimizer, our proof still correctly shows that for an arbitrary schedule A, there exists a schedule B such that Adam-with-schedule-B is equivalent to Momentum-with-schedule-A. But we do not believe it is useful to consider the learning rate schedule as “part of the algorithm” since, in practice, people use all kinds of different schedules with Adam (or any of the other popular algorithms).
> > > >
> > > > After I said that the bias correction term can be viewed as a learning rate schedule, you started to use it as an argument that then there is no difference since any learning rate schedule can be used. However, I meant that the bias correction term can be viewed as a learning schedule not that there is an exact translation. In fact, there is no as you can see in Algorithm 1 of https://arxiv.org/pdf/1412.6980.pdf where the two momentums also interplay with epsilon. Thus, it is only in approximation one can view it as a learning rate  schedule.
> > > > You don't just change the learning rate of Adam to endup with momentum SGD, you would also need to remove the bias correction term. If a practitioner would use some learning rate decay for Adam, then this decay would be on top of the bias correction effect and not instead of it.

---

> > > > > ### Author Response · Authors · 2019-11-15
> > > > > **We include the bias correction term as b_{t+1} in Table 1**
> > > > >
> > > > > As stated in the appendix, we use TensorFlow’s AdamOptimizer implementation, which differs trivially from the algorithm described in the Adam paper. Regardless, since our inclusion proof takes beta2=0, it applies equally well to both the TensorFlow AdamOptimizer and the algorithm in the Adam paper. We do not “drop the bias correction term” in either our experiments or our proof (see b_{t+1} in Table 1).
> > > > >
> > > > > In fact, the observation that Adam can approximate momentum is such an uncontroversial observation that it is included as an exercise in an undergraduate course at the University of Toronto (co-taught by Jimmy Ba, one of the authors of the original Adam paper), see Question 2(b) of http://www.cs.toronto.edu/~rgrosse/courses/csc421_2019/homeworks/hw2.pdf . We do not consider this observation to be a contribution of our paper; rather, our contribution is showing that this observation predicts relative optimizer performance under a realistic tuning protocol and budget.

---

> > > > > > ### Comment · AnonReviewer1 · 2019-11-15
> > > > > > **Re**
> > > > > >
> > > > > > >> In fact, the observation that Adam can approximate momentum is such an uncontroversial observation that it is included as an exercise in an undergraduate course at the University of Toronto (co-taught by Jimmy Ba, one of the authors of the original Adam paper)
> > > > > >
> > > > > > My first comment on the issue already clarified it well enough:
> > > > > >
> > > > > > "First, I would like to note that the claim that SGD with momentum is a special case of Adam with large epsilon is technically wrong because Adam also includes the bias-corrected momentum estimates which SGD with momentum does not consider. It might seem like a small difference, however it is a form of learning rate schedule which most users of Adam are not aware of. In practice, however, Adam with large epsilon can approximate SGD with momentum. Just don't claim the equivalent since it is not there."
> > > > > >
> > > > > > instead "Adam approximately equivalent to momentum SGD" as the course you mentioned says.

---

> > > > > > > ### Author Response · Authors · 2019-11-15
> > > > > > > **It seems we mostly agree**
> > > > > > >
> > > > > > > Our definition of optimizer inclusion (or specialization) was always: A includes B if A can approximate B arbitrarily well up to hyperparameter schedules. See Definition 1 for the formal statement. It seems that, under this definition, we are in agreement that Adam includes momentum (or momentum specializes Adam), since a learning rate schedule can be adjusted to remove bias correction for large epsilon.
> > > > > > >
> > > > > > > We disagree with the original statement in the review regarding the claims of our paper --- we never, in any revision or rebuttal, claimed that Adam can approximate momentum without a learning rate schedule adjustment.

---

> > > > > > > > ### Comment · AnonReviewer1 · 2019-11-15
> > > > > > > > **Re**
> > > > > > > >
> > > > > > > > For the sake of completeness I mention that the revised version includes
> > > > > > > >
> > > > > > > > "In particular, to approximate MOMENTUM with ADAM, one needs to choose a learning
> > > > > > > > rate schedule that accounts for ADAM’s bias correction."
> > > > > > > >
> > > > > > > > after my original note about it in the review.
> > > > > > > >
> > > > > > > > It is time stop this thread.

---

> ### Author Response · Authors · 2019-11-09
> **Clarifying the contribution of our paper (response to 2nd paragraph of review #1)**
>
> To facilitate discussion, we have responded to the three paragraphs of review 1 in separate threads. The 2nd paragraph seems to contain the reviewer’s primary concerns, so we focus on that first. Our latest revision should resolve the other issues mentioned by the review.
>
> - - - Summary - - -
>
> 1. We do not think the concerns raised in the 2nd paragraph are reasonable, nor is it reasonable to accuse us of acting in bad faith after we exceeded the standards of transparency and care in the literature (e.g. by reporting preliminary search spaces as well as final ones).
>
> 2. The reviewer was concerned that search spaces with nominally larger learning rate ranges for SGD and Momentum might be unfair relative to Adam, even though Adam faces a higher dimensional search problem.
> ● Although part of the point of our work is that we don't think any search spaces are completely fair, we do not think our original search spaces were unreasonable or biased our conclusions. Nevertheless, we ran additional experiments on CIFAR10 designed to, if anything, give non-adaptive optimizers an advantage and confirmed that using the "same" ranges wouldn't change our conclusions.
>
> 3. The reviewer was concerned that because we decided to search (epsilon, alpha0/epsilon) instead of searching (epsilon, alpha), we somehow unfairly penalized the non-adaptive optimizers because the former parameterization of the search space is more efficient for Adam in our experience.
> ● Our choice here is akin to log-transforming 1-momentum when tuning Momentum or log transforming learning rate. Only the adaptive optimizers *have* an epsilon parameter that needs to be searched, so this cannot be unfair to SGD or Momentum since they already benefit from not having to tune the hyperparameter at all.
>
> - - - Full response - - -
>
> One of the central points of our paper is that any empirical comparison of optimizers depends on the tuning protocol. No protocol we are aware of can guarantee fairness between optimizers with incommensurate search spaces -- and yet, empirical optimizer comparisons are crucial for developing new optimizers and guiding practitioners training neural networks. To our knowledge, no other study has highlighted how these comparisons are sensitive to the tuning protocol and which hyperparameters are tuned. Regarding our own results, we make clear in Section 5 that we should only expect our detailed findings to hold for similar workloads under similar protocols.
>
> Although it is always difficult to guarantee fairness when tuning over optimizers with incommensurate search spaces, this is still true for protocols that attempt to use the "same" search space for Adam and Momentum (Adam’s learning rate parameter is more closely related to (learning rate)/(1 - momentum), so the optimal ranges are almost always different between the two optimizers). Why should practitioners tie one hand behind their backs and only search a set of alpha values when tuning Adam that are the same as the set of learning rate values they search when tuning SGD? Given the importance of tuning protocols, practitioners must decide which protocol most closely captures their own when deciding which optimizer comparison is most relevant to them. Since our protocol produces results that exceed the performance from other optimizer comparisons in most cases (see Figures 3 and 4), we expect that readers will prefer using something similar to our tuning protocol.
>
> In light of reviewer 1’s concerns about our CIFAR-10 experiments, and to further support our claim that all optimizers were well tuned, we ran additional experiments with ResNet-32 on CIFAR-10. We ran an experiment with SGD, Momentum, and Nesterov with a search space where all learning rate and momentum ranges had the same width as the ranges for Adam's similar hyperparameters (ignoring that they are not necessarily the same units). We centered these new ranges on the best points from the original search, making the new comparison, if anything, unfair to the adaptive methods. Although further narrowing the search space about the best validation error reduces the mean validation error, our conclusions do not change from what was reported in the paper. Taking the process yet further, we doubled the trial budget for all optimizers and re-ran these equal-width-search-space experiments. Again, although the best validation error improved slightly for all optimizers, test error did not change much, and we see no clear evidence that the inclusion relationships are violated. See imgur.com/a/j38e1HP.
>
> Ultimately, it is extremely difficult to judge whether one of two incommensurate search spaces provides some sort of advantage -- again, a key point in our paper -- but we firmly believe our results are robust and that our protocol did not unrealistically bias our conclusions in favor of any one optimizer over another.

---

> > ### Comment · AnonReviewer1 · 2019-11-14
> > **Re**
> >
> > >> Only the adaptive optimizers *have* an epsilon parameter that needs to be searched, so this cannot be unfair to SGD or Momentum since they already benefit from not having to tune the hyperparameter at all.
> >
> > Yes, they have an epsilon and it is treated in a particular way. If you would optimize it without performing any transformation such as alpha0/epsilon you would get worse results and that would affect your conclusion. Now, you come up with a transformation and you get better results. What was the purpose of the transformation if not to get better results = if not to affect your conclusion about similarities in performance of Adam and momSGD.
> >
> > >> Taking the process yet further, we doubled the trial budget for all optimizers and re-ran these equal-width-search-space experiments.
> >
> > With the curse of dimensionality at hand and *when* the search space dimensionality is large, random search or grid search will happily consume 10x budget without deriving much.

---

> > > ### Author Response · Authors · 2019-11-15
> > > **Getting better results is a good thing, not a bad thing**
> > >
> > > The idea that adaptive gradient methods generalize worse than non-adaptive methods is a widely-held belief in our community. Our paper points out that this conclusion is theoretically unlikely as well as untrue under a protocol that anyone can implement.
> > >
> > > If tuning (epsilon, alpha0/epsilon) vs (epsilon, alpha) was the only reason we got good results with adaptive gradient methods and thus the only reason they performed better than non-adaptive methods, then we would be happy to present that as a major contribution of our paper since this is a simple trick that anyone can do. However, we find that the primary cause of the confusion around whether adaptive gradient methods perform worse than non-adaptive methods is whether or not epsilon is tuned, which is what we emphasize in the paper.
> > >
> > > Despite the reviewer’s skepticism about our search spaces and tuning protocol, our test errors (including Momentum’s and plain SGD's) are better than previous optimizer comparisons (Wilson et al., 2017, Schneider et al., 2019).

---

> > > > ### Comment · AnonReviewer1 · 2019-11-15
> > > > **Re**
> > > >
> > > > >> The idea that adaptive gradient methods generalize worse than non-adaptive methods is a widely-held belief in our community
> > > >
> > > > It was shown that it is partially due to the use of L2 regularization and not weight decay. The use of L2 and not weight decay is yet another thing that would make a difference between Adam and momentum SGD, i.e., Adam with L2 does not translate to SGD with L2 due to the adaptive part of Adam. You would need AdamW instead.
> > > >
> > > > >> If tuning (epsilon, alpha0/epsilon) vs (epsilon, alpha) was the only reason we got good results with adaptive gradient methods and thus the only reason they performed better than non-adaptive methods
> > > >
> > > > See the reply of Sachin Rajoria.
> > > >
> > > > >> Despite the reviewer’s skepticism about our search spaces and tuning protocol, our test errors (including Momentum’s and plain SGD's) are better than previous optimizer comparisons (Wilson et al., 2017, Schneider et al., 2019).
> > > >
> > > > Your target error rates for CIFAR-10 is 7% this is something that people used to show for ICLR 2015. WideResnets published in 2016 already had about 4% error in baseline settings. In other words, the regime that you show the results for is of little interest - small networks or networks not trained long enough.

---

> > > > > ### Author Response · Authors · 2019-11-15
> > > > > **Re**
> > > > >
> > > > > We also have experiments with ResNet-50 and Transformer which we do not consider models of “little interest - small networks”.

---

### Official Review · AnonReviewer3 · 2019-10-24
**Official Blind Review #3**

**Rating:** 6

**Review:**

The paper provides an empirical comparison of a set of first-order optimization methods for deep learning models. Those optimizers include stochastic gradient descent, momentum  method, RMSProp, Adam, Nesterov, and Nadam, which arguably covers all popular variants used in the literature. Although it is not the first empirical study on this topic, its conclusion differs slightly. The conclusion is a rather intuitive one: With proper parameter search, the 'richer', more powerful optimizers tend to work better, regardless of the downstream tasks.

Pros:
- Intuitive results with a well designed workloads and experiments. For practitioners that want to start their own hyperparameter search, the workloads and setups are likely to be useful.

Cons:
- I am not entirely convinced that the inclusion relationship is indeed a major cause or indicator of different optimizers' performance. There is no theoretical justification; Empirically, if one takes two optimizers equally rich and tunes one of them more intensively, one should expect a better performance, too.

Suggestions:

- I think at least the basic definitions of different optimizers should be given in the main text. Otherwise, readers without detailed knowledge of all these optimizers cannot follow the paper. For example, the paper starts talking about the taxonomy of the optimizers with their corresponding hyperparameters in Section 3.2 before giving any functional form of the optimizers.

- I would suggest the authors to follow the convention and use the term "hyperparameter" rather than "metaparameter". The readers of this paper are not primarily Bayesian, there is really no need to divert from the convention. Besides, the term "Bayesian hyperparameter tuning" is widely used even.

- I wonder to which extent the network structures impact the choice of the hyperparameter (e.g., CNN vs. RNN).

**Experience Assessment:**

I have read many papers in this area.

**Review Assessment: Checking Correctness Of Derivations And Theory:**

N/A

**Review Assessment: Checking Correctness Of Experiments:**

I assessed the sensibility of the experiments.

**Review Assessment: Thoroughness In Paper Reading:**

I read the paper at least twice and used my best judgement in assessing the paper.

---

> ### Author Response · Authors · 2019-11-09
> **We have incorporated these suggestions into the latest version**
>
> Thank you for the review. We believe we have incorporated all suggestions into the latest revision of the manuscript. Regarding the effect of network structure on hyperparameter choices, we agree that this is an important point. All aspects of the workload likely affect the best hyperparameter configurations, at least to some extent. It is an interesting question whether there is more structure in these effects, and we leave that question to future work.
>
> Regarding the inclusion relationships, it is certainly true that tuning, say, RMSProp much more thoroughly than Adam will make RMSProp get better results if neither optimizer has been close to optimally tuned. But as we tune all optimizers more and more carefully, the theoretical inclusion relationships will become the dominant effect. That still doesn't tell us what will happen between RMSProp and Adam since they don't include each other, but it does tell us what will happen between Momentum and Adam. At some point, given a particular family of learning rate schedules, it will no longer be possible to improve Momentum with additional tuning (at least on the test set; we can overfit the validation set just by trying more and more random seeds).
>
> Although a crucial point of our paper is that tuning protocols matter a lot and there may not be a way to be completely fair when comparing different optimizers, we are not saying that nothing can be learned from empirical comparisons. If the reader is willing to accept our particular parameterization of the learning rate schedules, we believe our conclusions will not change as we use more and more tuning trials in our setup. Indeed, the results of our additional experiments in response to reviewer #1 show that although we can continue to reduce validation error slightly by narrowing our search spaces and/or running more trials, we cannot reduce our test error for ResNet-32 on CIFAR-10 with more tuning, and our conclusions remain the same regardless.
>
> We hope that we have adequately addressed all the concerns in this review and that the reviewer will consider raising their score accordingly. Please let us know if you believe additional issues remain with the newest version.

---

### Public Comment · ~Boris_Ginsburg1 · 2019-09-30
**Is Adam with large epsilon still Adam?**

The epsilon values used fin Adam and Nadam  are very large . For example epsilon used in ResNet-50 experiments are [10^-2; 10^2] and for Nadam [10^3;10^7]. Such high epsilon effectively disable the normalization of 1st moment by second moment, and algorithm becomes as SGD with momentum. This clearly contradicts to the whole idea beyond Adam.  Can you still call it Adam?

---

> ### Public Comment · ~Liyuan_Liu2 · 2019-09-30
> **On the epsilon value**
>
> Good question, I have the same doubt for a long time...
>
> Personally speaking, setting epsilon to a large value makes the resulting algorithm different from the vanilla Adam. Intuitively, its effect is the same with calculating the arithmetic average between the raw 2rd moment and a prior. If the arithmetic average is changed to be the geometric average, the resulting algorithm is PAdam [1]. Therefore, Adam-with-large-epsilon is more like a variant of Adam.
>
> At the same time, in some cases, people refers 'Adam with a large epsilon' as 'Adam' (e.g., it seems that using a large epsilon is quite common & useful in reinforcement learning). In these cases, it is not treated as a variant...
>
> Overall, I am not sure whether we should refer 'Adam-with-large-epsilon' as a new variant, or as 'Adam' after parameter tuning.
>
> Chen, Jinghui, and Quanquan Gu. "Closing the generalization gap of adaptive gradient methods in training deep neural networks." arXiv preprint arXiv:1806.06763 (2018).

---

> > ### Public Comment · ~Boris_Ginsburg1 · 2019-10-05
> > **E-Adam?**
> >
> > Maybe the paper novelty is in suggesting a new  way how you can morph Adam into SGD with momentum? For example you can start training  with small epsilon (Adam ) and increase epsilon during training  to 100 (SGD), similarly to Padam?

---

> > > ### Public Comment · ~Liyuan_Liu2 · 2019-10-06
> > > **Adam-with-large-epsilon is not novel but it may be the first time to have the detailed comparison**
> > >
> > > Hmmm, personally, I think the major contribution of this paper, as suggested by the title, is the empirical comparison. Since setting a larger epsilon is briefly mentioned in existing literature (e.g., in the doc for f.train.AdamOptimizer), I don't think the author will choose to claim this as the novelty. Still, to the best of my knowledge, it is the first time to have a detailed empirical study on the effect of this parameter.

---

> > > > ### Public Comment · ~Sachin_Rajoria2 · 2019-10-21
> > > > **Previous detailed comparison**
> > > >
> > > > Previously a more thorough comparison in my opinion has been carried out in https://papers.nips.cc/paper/8186-adaptive-methods-for-nonconvex-optimization.pdf and appendix D: https://papers.nips.cc/paper/8186-adaptive-methods-for-nonconvex-optimization-supplemental.zip
> > > >
> > > > Across multiple domains and models, large epsilon (1e-8) has been compared to default epsilon (1e-8) in Adam. Another interesting thing noted was often use of large epsilon slowed down initial progress, so one should do super quick early termination when doing parameter search.

---

> > > > > ### Comment · AnonReviewer1 · 2019-10-21
> > > > > **typo**
> > > > >
> > > > > "large epsilon (1e-8)" ->"large epsilon (1e-3)"

---

> > > > > ### Public Comment · ~Liyuan_Liu2 · 2019-11-05
> > > > > **Thanks for the reference : -)**
> > > > >
> > > > > Thanks for pointing out the reference : -)

---

> ### Author Response · Authors · 2019-10-01
> **"A rose by any other name would smell as sweet"**
>
> We agree that there are some subtle questions here. First, and most important from our perspective, should researchers tune epsilon over ranges that include large epsilon values? Our results strongly suggest the answer is yes! We found that starting with Adam and tuning all its metaparameters is an effective way to proceed. On some workloads, a larger epsilon is optimal, while on other workloads, like Transformer, a smaller epsilon is optimal. Since we can't predict the best value of epsilon ahead of time, we need to tune it. Our related work section mentions several examples where other researchers have found relatively large values of epsilon to be useful, and as Liyuan Liu points out, larger values are common in reinforcement learning.
>
> Second, can we still call Adam with very large values of epsilon “Adam”? This is an inherently difficult question to answer. As we show in Appendix A (and as you also pointed out), Adam continuously approximates Momentum in the limit as epsilon approaches infinity. At which finite value should Adam be referred to as Momentum? This should depend on the objective function being optimized. In our paper, we refer to the algorithm from Kingma and Ba (2015) with any finite positive epsilon as “Adam”.

---

> > ### Public Comment · ~Boris_Ginsburg1 · 2019-10-05
> > **"Adam. What’s in a name?"**
> >
> > Adam is fundamentally based on the idea of adaptation of first moment by  second moment estimation, and the original motivation for epsilon was just to avoid dividing by zero. It would be very interesting to see how large is the value of second moment comparing to epsilon, especially  when  you use large epsilon. From hyper-parameter search point of view, large epsilon unify Adam and SGD with momentum into one search. But I wonder, if there is a gain in checking whole epsilon range vs just doing hyper-parameter search for Adam and SGD separately?

---

### Public Comment · ~Matthias_Minderer1 · 2019-10-16
**Hyperparameters vs metaparameters?**

Why do you use the term "metaparameter" instead of the more common "hyperparameter"?

I'm not familiar with the term "metaparameter", but Wiktionary suggests that a metaparameter is a "parameter that controls the value of one or more others", whereas hyperparameter is "a parameter whose value is set before the learning process begins". The latter seems to be more generally appropriate for the parameters discussed in the paper, even though some of them may qualify as metaparameters.

If the distinction is important, it might be worthwhile to comment on it in the paper (sorry if you already do and I missed it). If it's not important, it may be best to stick with the more common term to avoid confusion.

---

> ### Author Response · Authors · 2019-10-17
> **"Hyperparameter" is not the most technically correct term**
>
> We use the term "metaparameter" because "hyperparameter" has a specific technical meaning in Bayesian machine learning as a parameter that controls a prior distribution over other parameters. Although admittedly a pedantic point, most uses of "hyperparameter" in the deep learning literature to describe quantities such as learning rates are technically incorrect. We believe the Bayesian usage of the term should prevail here and we should use the more generic term "metaparameter" to refer to all the configuration parameters and optimizer parameters we tune using validation error when training neural networks. Radford Neal and other prominent Bayesian machine learning researchers have made this complaint for years.

---

> > ### Comment · AnonReviewer1 · 2019-10-17
> > **but**
> >
> > Metaparameters were popularized in the evolution computation community in the 80ies for defining and optimizing parameters of GAs/EAs. This was before MacKay et al. popularized hyperparameters in the early 90ies. I don't think that the use of the term "hyperparameter" outside of its original domain takes away anything from the Bayesian community. Since thousands of papers in ML and optimization define things like "all the configuration parameters and optimizer parameters we tune using validation error when training neural networks" as hyperparameters, it might be a little bit too late and more confusing than useful to redefine back as metaparameters.

---

### Public Comment · ~Frank_Schneider1 · 2019-10-22
**Regarding DeepOBS**

Full disclosure: We are the authors of DeepOBS [Schneider et al., ICLR 2019]. We are _not_ assigned as reviewers to this paper. Since this paper contains several statements about our work, we nevertheless would like to clarify some aspects of our work to the benefit of the community and the reviewers.

We built DeepOBS as a tool for researchers who develop new deep learning optimizers, to allow them to efficiently test their new method on realistic but feasible architectures and compare to baselines. DeepOBS offers a set of test problems, benchmarks, and automated evaluation procedures. One of our goals was to encourage such authors to make their new methods practically useful. We thus also chose the benchmarks to reflect the kind of effort an applied end-user of deep learning might realistically invest in parameter-tuning.

To be clear: It was never a core goal of DeepOBS to argue in favor or against any particular optimizer. The DeepOBS paper only offered benchmarks on SGD, Adam and momentum SGD, as these remain among the most popular ones among practitioners. We expected all three of these to be beaten handily by newer methods (not because they are bad, but because newer methods aren’t interesting if they don’t even beat the most common competitors). In fact, everyone is invited to contribute their own optimizers to the DeepOBS benchmark, we happily accept pull-requests to
our repo at https://deepobs.github.io and add such results to the leaderboard.

It is correct that for our analysis, we treated the $\beta_1$, $\beta_2$, and $\epsilon$ parameters of Adam as constant because, as the present authors note themselves, this is common practice. The present paper argues that Adam should always dominate (SGD) because, when all these are treated as free parameters, Adam actually contains (momentum) SGD as a special case (Adam then has 4 parameters, compared to SGD's single one). This is an interesting point. If it turns out that these four parameters can indeed be tuned as efficiently as the paper argues (i.e. that the same search budget for SGD and 4-parameter-Adam leads to better performance for the latter), we would be very happy to add this (arguably new) optimizer to our benchmark.

One problem we see in this regard is that the present paper pre-specifies different search domains for each of the benchmark problems. For example, for FashionMNIST, the authors decided to use very small (i.e. the usual) values for Adam’s epsilon (Table 7), but for ImageNet they searched only large values of epsilon (Table 19) (in that latter case, the update rule of Adam is arguably closer to momentum SGD than to Kingma’s & Ba’s original Adam). So the paper actually labels a different algorithm as `Adam’ for each separate test problem. The authors explain these choices in Section 4 by stating that

> […] the provenance of each search space is difficult to trace exactly. In some cases, our search spaces were informed by published results or prior experience with particular models and optimizers.

This stance is difficult to reconcile with the notion of a benchmark. We believe a benchmark should be indicative of the performance the evaluated method would have on _new_, not previously explored problems, because this is the situation practitioners actually face. To the user, an optimizer consists of an update rule and a space of (not just a number of) tunable hyperparameters. If the space of good hyperparameters differs from one benchmark to the next then, for a new problem, the true search-space really is the union of all these spaces. How would a user otherwise know how to choose the search space on a domain-model-combination they have never faced before?

We want to emphasize that we very much welcome this work, including its constructive criticism of DeepOBS, because we strongly believe that the current practice of empirical comparisons of deep learning optimizers is flawed. While DeepOBS certainly does not solve all problems, we think that it is a good step in the right direction. We would be very happy to work with the authors to see whether their insights could make their way into a future version of DeepOBS.

---

### Decision · Program_Chairs · 2019-12-19

**Decision:**

Reject

**Comment:**

This paper examines classifiers and challenges a (somewhat widely held) assumption that adaptive gradient methods underperform simpler methods.

This paper sparked a *large* amount of discussion, more than any other paper in my area. It was also somewhat controversial.

After reading the discussion and paper itself, on one hand I think this makes a valuable contribution to the community. It points out a (near-) inclusion relationship between many adaptive gradient methods and standard SGD-style methods, and points out that rather obviously if a particular method is included by a more general method, the more general method will never be worse and often will be better if hyperparameters are set appropriately.

However, there were several concerns raised with the paper. For example, reviewer 1 pointed out that in order for Adam to include Momentum-based SGD, it must follow a specialized learning rate schedule that is not used with Adam in practice. This is pointed out in the paper, but I think it could be even more clear. For example, in the intro "For example, ADAM (Kingma and Ba, 2015) and RMSPROP (Tieleman and Hinton, 2012) can approximately simulate MOMENTUM (Polyak, 1964) if the ε term in the denominator of their parameter updates is allowed to grow very large." does not make any mention of the specialized learning rate schedule.

Second, Reviewer 1 was concerned with the fact that the paper does not clearly qualify that the conclusion that more complicated optimization schedules do better depends on extensive hyperparameter search. This fact somewhat weakens one of the main points of the paper.

I feel that this paper is very much on the borderline, but cannot strongly recommend acceptance. I hope that the authors take the above notes, as well as the reviewers' other comments into account seriously and try to reflect them in a revised version of the paper.